# NFATc2 enhances tumor-initiating phenotypes through the NFATc2/SOX2/ALDH axis in lung adenocarcinoma

**Zhi-Jie Xiao[1], Jing Liu[1], Si-Qi Wang[1], Yun Zhu[1], Xu-Yuan Gao[1], Vicky Pui-Chi Tin[1], Jing Qin[2], Jun-Wen Wang[3,4], Maria Pik Wong[1]\***

[1]Department of Pathology, The University of Hong Kong, Hong Kong, Hong Kong; [2]School of Life Sciences, The Chinese University of Hong Kong, Shatin, Hong Kong; [3]Department of Health Sciences Research AND Center for Individualized Medicine, Mayo Clinic, Scottsdale, United States; [4]Department of Biomedical Informatics, Arizona State University, Scottsdale, United States

**Abstract** Tumor-initiating cells (TIC) are dynamic cancer cell subsets that display enhanced tumor functions and resilience to treatment but the mechanism of TIC induction or maintenance in lung cancer is not fully understood. In this study, we show the calcium pathway transcription factor NFATc2 is a novel regulator of lung TIC phenotypes, including tumorspheres, cell motility, tumorigenesis, as well as in vitro and in vivo responses to chemotherapy and targeted therapy. In human lung cancers, high NFATc2 expression predicted poor tumor differentiation, adverse recurrence-free and cancer-specific overall survivals. Mechanistic investigations identified NFATc2 response elements in the 3' enhancer region of *SOX2*, and NFATc2/SOX2 coupling upregulates ALDH1A1 by binding to its 5' enhancer. Through this axis, oxidative stress induced by cancer drug treatment is attenuated, leading to increased resistance in a mutation-independent manner. Targeting this axis provides a novel approach for the long-term treatment of lung cancer through TIC elimination.

\*For correspondence: mwpik@hku.hk

**Competing interests:** The authors declare that no competing interests exist.

## Introduction

Lung cancer results from mutations induced by DNA adducts, free radicals and reactive oxygen species (ROS) generated during tobacco smoking and chronic inflammation (*Acharya et al., 2010*; *Okumura et al., 2012*; *Houghton, 2013*). Due to late presentation and the lack of effective long term therapy, it has a high mortality and new treatment approaches are needed to improve patient outcomes. Recent research has shown the cellular landscape of a cancer is heterogeneous. Cells showing aberrant expressions of various molecules have enhanced propensities for survival, tumorigenicity, drug resistance, designated as cancer stem cells or tumor-initiating cells (TIC). In some cancers, constitutive activities of inherent embryonic or developmental pathways for stem cell renewal are involved in TIC maintenance, such as the WNT/$\beta$-catenin pathway in colonic adenocarcinomas (AD). In tissues with slow cell turnover, mechanisms that elicit cell plasticity and stemness properties during tissue response to intracellular and extracellular stresses are involved (*Valent et al., 2012*; *Visvader and Lindeman, 2012*; *Beck and Blanpain, 2013*). For the adult lung, stem cell niches or their physiological regulatory mechanisms are ill-defined, and molecular programs sustaining lung TIC are still elusive.

Intracellular free calcium is at the hub of multiple interacting pathways activated by extracellular and/or intrinsic stimulations, e.g. EGFR, endoplasmic reticulum and mitochondrial stresses, etc. (*Roderick and Cook, 2008*; *Prevarskaya et al., 2010*; *Zhao et al., 2013*; *Déliot and Constantin,*

**eLife digest** Cancer develops when cells become faulty and start to grow uncontrollably. They eventually form lumps or tumors, which may spread to surrounding tissues or even to other areas in the body. One of the reasons why cancer treatment remains a challenge is that there are over 200 types of cells in the body, and there are a lot of moments in the life cycle of a cell when things could go wrong. Researchers have shown that many cancers, including lung cancer, are not only extremely different from patient to patient, but also display great differences between cancer cells within the same tumor.

Increasing evidence suggest that these differences may be caused by a type of cells called tumor initiating cells, or TICs for short. These TICs behave like stem cells and can renew themselves or mature into different types of cells. They are thought to help cancers grow and spread, and even make them resistant to treatments. Previous research has shown that in many types of cancer, the protein NFATc2 helps cancer cells to grow and spread. Until now, however, it was not known if NFATc2 is also important in TICs in lung cancer.

Using human lung cancer cell lines and animal models, Xiao et al. show that the protein NFATc2 stimulates the stem-cell like behavior of TICs. The results showed that TICs had higher levels of the NFATc2 protein than other lung cancer cells that were not TICs. Tumors with higher levels were also more aggressive. When NFATc2 was removed from the cells, they formed smaller tumors and were more sensitive to drug treatment compared to cancer cells with NFATc2.

Further experiments revealed that NFATc2 helped to increase the levels of a protein called Sox2, which gives cells the ability to renew or develop into different cell types. Together, these two proteins stimulated the production of another protein that was already known to play a crucial role in TIC maintenance.

A better understanding of the mechanisms regulating TICs in lung cancer will help scientists tackle new questions about how this cancer progresses and resists to therapy. In the longer-term, combining classic cancer treatments with new therapeutic strategies targeting NFATc2 could make treatments for lung cancer patients more effective.

*2015*), raising the possibility stress signals transduced by calcium pathway mediators could be involved in the induction of TIC phenotypes. In cancers of the breast, pancreas, colon and melanoma, the calcium signaling transcription factor nuclear factor of activated T-cells (NFAT) has been shown to contribute to malignant properties including cell invasion, migration, survival, proliferation, stromal modulation and angiogenesis (*Werneck et al., 2011*; *Gerlach et al., 2012*; *Qin et al., 2014*). However, information on its role in TIC phenotypes, especially drug resistance, is limited and details of the molecular pathways linking calcium signaling to TIC induction have not been reported. In this study, we demonstrated the parlor NFATc2 supports tumorigenicity, cell survival, motility and drug resistance of human lung AD. Amongst essential factors of pluripotency, SOX2 is an NFATc2 target upregulated through its 3' enhancer, while SOX2 couples to a 5' enhancer of *ALDH1A1* mediating overexpression. NFATc2 induces ALDH$^+$ subsets and ALDH$^+$/CD44$^+$-TIC and enhances drug resistance by ROS scavenging through the NFATc2/SOX2/ALDH1A1 axis. Our study reports a novel lung TIC maintenance pathway which links micro-environmental stimulation to induction of stemness phenotypes, evasion of cell death and enhancement of drug resistance. NFATc2 could be an important target in treatment strategies aiming at disruption of lung TIC.

## Results

### NFATc2 expression correlated with adverse survivals of human NSCLC

By analyzing transcripts expression, we observed *NFATc2* was significantly overexpressed in human primary NSCLC compared to normal lung (*Figure 1A*). Using IHC, high level activated NFATc2 expression with intense and widespread nuclear staining were detected in 41 of 102 (40.2%) excised primary NSCLC, while 61 (59.8%) showed low expression with weak nuclear and/or cytoplasmic staining in isolated or small clusters of tumor cells (*Figure 1B,C*). In normal lung epithelium, NFATc2

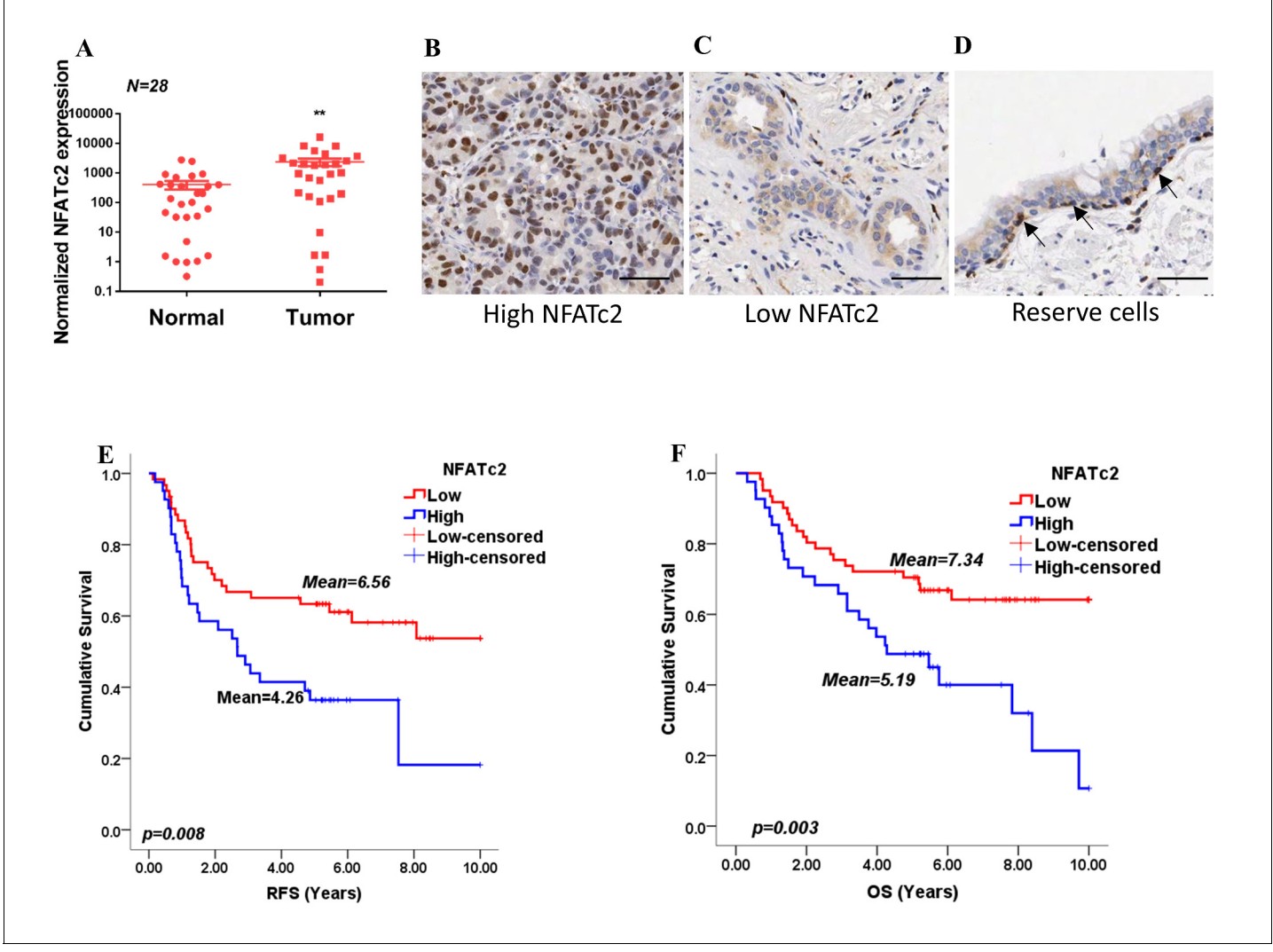

**Figure 1.** NFATc2 was overexpressed in human NSCLC and predicted poor survivals. (A) *NFATc2* expression analyzed by qPCR in human NSCLC and corresponding normal lung. P: Wilcoxon test. p=0.0003. (B–C) NFATc2 expression analyzed by IHC, showing representative areas of high NFATc2 scores with strong nuclear staining in the majority of cancer cells (B), or low NFATc2 scores with weak nuclear and cytoplasmic staining (C), respectively. (D) NFATc2 expression in normal bronchial epithelium by IHC, showing nuclear NFATc2 staining in scattered bronchiolar reserve/stem cells of the basal layer (arrows). For B-D: Scale bars, 50 μm. (E–F) Kaplan Meier survival curves by log-rank tests on 102 resected primary NSCLC stratified by NFATc2 expression levels for recurrence-free survival (RFS) (E), and overall survival (OS) (F).

The following source data is available for figure 1:

**Source data 1.** Statistical analyses for *Figure 1A*.

was expressed in the bronchiolar stem cell compartment of basal reserve cells while differentiated bronchiolar cells or alveolar pneumocytes were negative (*Figure 1D*). Using log rank test and Kaplan-Meier survival analysis, we showed tumors with high level NFATc2 expression had significantly shorter recurrence-free survival (RFS) and cancer-specific overall survival (OS) (*Figure 1E,F*). High NFATc2 expression significantly predicted poor tumor differentiation, advanced tumor stage and TNM stage (*Table 1A*). Multivariate Cox regression analysis further showed high NFATc2, late pathological stage, age and smoking history were independent prognostic indicators for shorter OS, while high NFATc2 and advanced pathological stage were predictive for shorter RFS (*Table 1B, C*). The results indicated NFATc2 expression was associated with repressed tumor differentiation and adverse patient survivals.

**Table 1.** Clinico-pathological correlation of NFATc2 in NSCLC patients.

**A. Clinico-pathological correlation of NFATc2 in NSCLC patients**

| | NFATc2 | | |
|---|---|---|---|
| Clinico-pathological variables | Low | High | P value |
| Gender | | | |
| Female | 22 | 11 | 0.328 |
| Male | 39 | 30 | |
| Age (Years) | | | |
| ≤65 | 36 | 24 | 0.961 |
| >65 | 25 | 17 | |
| Smoking history | | | |
| Non-smoker | 32 | 24 | 0.545 |
| Smoker | 29 | 17 | |
| Differentiation | | | |
| Well to moderate | 45 | 21 | 0.019* |
| Poor | 16 | 20 | |
| Histologic type | | | |
| Adenocarcinoma | 42 | 24 | 0.357 |
| Squamous cell carcinoma | 11 | 10 | |
| Others | 8 | 7 | |
| Tumor Stage | | | |
| T1-T2 | 53 | 24 | 0.001* |
| T3-T4 | 8 | 17 | |
| Lymph node metastasis | | | |
| Absent | 43 | 25 | 0.318 |
| Present | 18 | 16 | |
| Pathological (TNM) stage | | | |
| Stage I | 36 | 14 | 0.014* |
| Stage II-IV | 25 | 27 | |

**B. Multivariate COX regression analysis for RFS**

| Variables | P value | Hazard Ratio (HR) | 95.0% CI[†] of HR |
|---|---|---|---|
| NFATc2 | 0.037 | 1.905 | 1.039–3.494 |
| TNM stage | 0.001 | 2.035 | 1.347–3.075 |

**C. Multivariate COX regression analysis for OS**

| Variables | P value | Hazard ratio (HR) | 95.0% CI[†] of HR |
|---|---|---|---|
| NFATc2 | 0.002 | 2.824 | 1.462–5.457 |
| TNM stage | 0.012 | 1.827 | 1.140–2.927 |
| Age | 0.01 | 2.33 | 1.224–4.432 |
| Smoking history | 0.009 | 2.416 | 1.251–4.665 |

Statistical tests: $c^2$; *: P<0.05

†: Confidence Interval.

Statistics: COX regression analysis.

## NFATc2 was overexpressed in lung TIC and mediated TIC properties

If NFATc2 supports tumor-initiating phenotypes, it is expected to be expressed at a higher level in TIC compared to non-TIC. To address this notion, marker-independent TIC surrogates comprising tumorspheres raised from 4 lung AD cell lines cultured in non-adherent, stem cell conditions (*Liu et al., 2007*; *Shi et al., 2015*; *Sun et al., 2015*) were compared with non-TIC growing in monolayers. TIC showed higher NFATc2 expression by Western blot (*Figure 2A*), while transcripts of *NFATc2* and its target *FASL* were also significantly upregulated (*Figure 2—figure supplement 1A*). Luciferase reporter assays also showed significantly higher NFAT activities in spheres isolated from H1299 and A549 cells (*Figure 2—figure supplement 1B*). Furthermore, TIC selected by the lung TIC markers ALDH$^+$/CD44$^+$ from HCC827 and the patient-derived lung cancer cell lines, HKUCL2 and HKUCL4, showed higher NFATc2 expression than the ALDH$^-$/CD44$^-$ non-TIC counterpart (*Figure 2B*)(*Liu et al., 2013a*). Using another lung TIC marker, CD166$^{high}$, for TIC isolation from HCC827, NFATc2 was also shown to be upregulated (*Figure 2C*) (*Zhang et al., 2012*).

For functional studies, NFATc2 was silenced by 2 shRNA sequences (shNFATc2-A and -B) in 2 lung cancer cell lines with high basal expression (HCC827, PDCL#24), and ectopically expressed in 2 cell lines with relatively low de novo expression (A549, H1299) (*Figure 2D*). Knockout using CRISPR/CAS9 and gRNA targeting NFATc2 (gNFATc2) was also performed on HCC827 (*Figure 2D*). As shown by BrdU proliferation assay and cell cycle assay, NFATc2 knockdown did not significantly affect the proliferation and cell cycle distribution of HCC827 cells (*Figure 2E and F*, *Figure 2—figure supplement 2*). However, abrogation of NFATc2 significantly reduced 60–70% of tumorspheres in all the cell models and inhibited tumorspheres renewability for 2 consecutive generations (*Figure 2G–I*), while overexpression significantly augmented tumorspheres in both A549 and H1299 (*Figure 2J–K*). To demonstrate the actions of NFAT on TIC-related phenotypes are mediated through calcium signaling, we blocked calcium-mediated NFAT activation by disrupting the upstream calcineurin/NFAT dephosphorylation complex using the potent and specific inhibitors cyclosporin A (CSA) and FK506, respectively. Treatment with both CSA and FK506 significantly inhibited sphere formation of HCC827 cells (*Figure 2L*). Transwell assays for cell motility showed silencing NFATc2 significantly reduced the migration and invasion ability of both HCC827 and PDCL#24 cells (*Figure 2M–N*), while the opposite effects were rendered by NFATc2 overexpression in A549 and H1299 cells (*Figure 2O–P*).

## NFATc2- mediated tumorigenesis in vivo

Subcutaneous xenograft models showed NFATc2 knockdown significantly reduced tumor sizes and retarded growth rates of HCC827 and PDCL#24, respectively (*Figure 3A–B*). Using IHC, NFATc2 expression was detected in xenografts established from HCC827 and PDCL#24 knockdown cells (*Figure 3C*), indicating successful tumorigenesis preferentially involved cells that had selected away from NFATc2 knockdown. In contrast, NFATc2 overexpression in A549 significantly augmented tumor growth (*Figure 3D*). To evaluate TIC frequencies *in vivo*, limiting dilution assays were performed by subcutaneous transplantation of serially decreasing numbers of tumor cells in nude mice. NFATc2 knockdown led to significantly reduced xenograft incidence and TIC frequency of HCC827 (*Figure 3E*, *Figure 3—figure supplement 1A*). Reciprocal effects were observed with NFATc2-overexpression, (*Figure 3F*, *Figure 3—figure supplement 1B*). Together, the data supported NFATc2-mediated *in vivo* tumorigenesis.

## NFATc2 promoted cancer resistance to cytotoxic and targeted therapy

The effect of NFATc2 on chemotherapy response was first investigated using cisplatin chemotherapy on PDCL#24, a *KRAS V12D* mutant lung AD cell line raised from a local male chronic smoker. Upon NFATc2 inhibition, sensitization to cisplatin with statistically significant reduction of IC$_{50}$ was observed compared to control cells (*Figure 4A*). Similarly, NFATc2 knockout as well as CSA and FK506 treatment also sensitized HCC827 to cisplatin with reduction of IC$_{50}$ (p<0.01) (*Figure 4—figure supplement 1A*, *Figure 4B*). In contrast, overexpressing NFATc2 in A549 increased cisplatin resistance with significantly elevated IC$_{50}$ (p<0.01) (*Figure 4C*). *In vivo*, mice bearing PDCL#24 xenografts treated with cisplatin alone showed 1.68 fold tumor shrinkage compared with vehicle control. With additional stable NFATc2 knockdown, tumor shrinkage was enhanced to 3.15 and 2.2 fold relative to respective vehicle controls (p<0.01) (*Figure 4D*), while tumor growth rate was also retarded

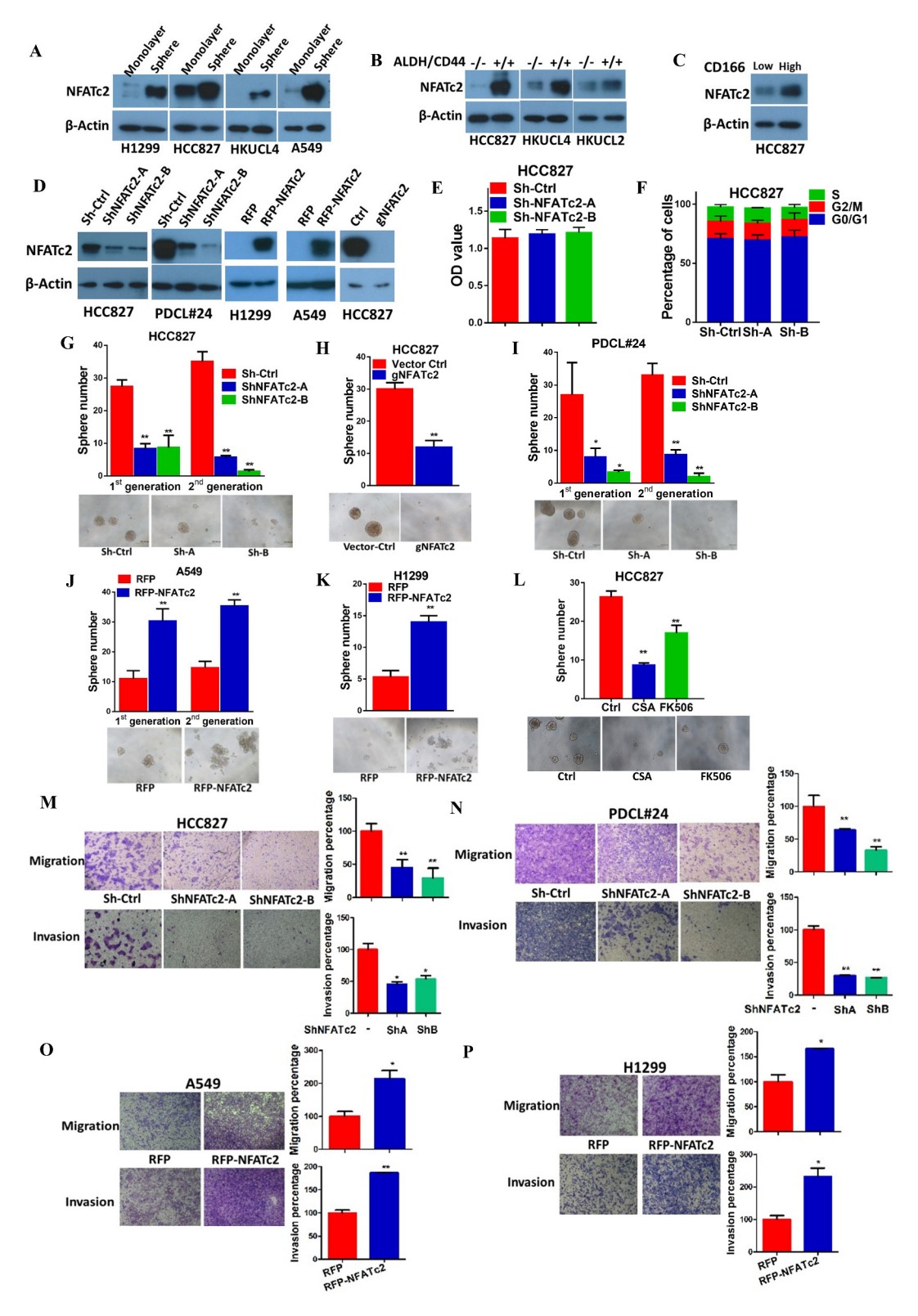

**Figure 2.** NFATc2 NFATc2 was overexpressed in lung TIC and mediated TIC properties in vitro. (A–C) Expression of NFATc2 analyzed by Western blot, in TIC isolated as tumorspheres compared with non-TIC from cells in monolayers (A); TIC isolated as ALDH[+]/CD44[+] subset compared with ALDH[-]/CD44[-] subset (B); TIC isolated as CD166[high] subset compared with the CD166[low] subset (C). (D) NFATc2 expression by Western blot in cells with stable NFATc2 knockdown, overexpression, or knockout, respectively. (E) BrdU proliferation assay of HCC827 cells with NFATc2 knockdown. (F) Cell cycle

*Figure 2 continued on next page*

Figure 2 continued

analysis of HCC827 cells with NFATc2 knockdown. (G–I) Tumorsphere formation and serial passage assays, in HCC827 cells after stable NFATc2 knockdown (G) or knockout (H), or in PDCL#24 cells with NFATc2 knockdown (I). (J–K) Tumorsphere formation and serial passage assays in cells with stable NFATc2 over-expression, including A549 cells (J) and H1299 cells (K). (L) Tumorsphere formation assay in HCC827 cells with or without treatment with 1 μM of CSA or 5 μM of FK506. (M–P) Cell migration and invasion assays in cells with stable NFATc2 knock down (M and N) or over-expression (O and P). For G-P: *p<0.05 **p<0.01, comparison with control by t-test. Error bar indicates the mean ±SD for at least three independent replicates.

The following figure supplements are available for figure 2:

**Figure supplement 1.** NFATc2 was up-regulated in tumorspheres.

**Figure supplement 2.** NFATc2 knockdown did not affect cell cycle progression of HCC827 cells.

(*Figure 4—figure supplement 1B*). Besides cisplatin, NFATc2 knockdown significantly increased paclitaxel sensitivity of HCC827 cells (*Figure 4—figure supplement 2A*). A549 cells induced for cisplatin resistance by chronic progressive drug exposure (A549 CR) showed elevated $IC_{50}$ with NFATc2 upregulation compared to parental A549 cells (*Figure 4E–F*). With NFATc2 suppression, A549 CR was re-sensitized with return of $IC_{50}$ to around the pre-induction level (p<0.01) (*Figure 4F*). In line with this observation, induction of H1299 for paclitaxel resistance also caused NFATc2 upregulation (*Figure 4—figure supplement 2B*).

HCC827 is a lung AD cell line known to harbor an activating *EGFR* exon19 deletion which sensitizes it to tyrosine kinase inhibitor (TKI) therapy. To investigate whether NFATc2 contributes to targeted therapy resistance, NFATc2 was stably inhibited by shRNA knockdown or CRISPR knockout. This led to significantly reduced $IC_{50}$ for the TKI gefitinib (*Figure 4G,H*). Similarly, co-treatment with either CSA or FK506, respectively, significantly sensitized HCC827 cells to gefitinib (*Figure 4I*). In vivo, although both groups of NFATc2 knockdown mice showed higher folds of tumor shrinkage (3.57 fold, 4.73 fold, respectively) after 2 weeks of gefitinib treatment compared to scramble control (3.38 fold) (*Figure 4J*), the added effects of NFATc2 inhibition were very modest and the effects on the growth curve was not clear-cut (*Figure 4—figure supplement 3A*). Using an alternative model, we investigated the effects of NFATc2 inhibition on response to short term gefitinib treatment for 5 days which allowed tumor recovery from the pronounced effect of gefitinib. In mice with NFATc2 knockdown, tumor regrowth was observed in only 3 and 2 mice of the sh-NFATc2-A and sh-NFATc2-B groups, respectively. In contrast, all mice in the control group showed tumor regrowth. The NFATc2-inhibited xenografts showed more effective tumor inhibition (5.44 fold, 39.73 fold, respectively) compared to the control group (3.54 fold) (p<0.01) and differences in tumor volumes on the growth curve were statistically significant (*Figure 4—figure supplement 3B–C*).

In HCC827 induced for gefitinib resistance (HCC827 GR), NFATc2 was upregulated and NFAT promoter activities were increased compared to parental cells (*Figure 4K–L*). Upon CSA inhibition or sh-NFATc2 knockdown, respectively, re-sensitization to gefitinib with significantly reduced $IC_{50}$ resulted (*Figure 4M*). Integrating the in vitro and in vivo data of various combinations of multiple cell lines and cancer drug treatments, the enhancing effect of NFATc2 on drug resistance to cytotoxic and targeted therapy was demonstrated.

## NFATc2 upregulated SOX2 expression through its 3' enhancer

To understand the molecular mechanism through which NFATc2 mediates cancer cell stemness and drug resistance, we hypothesize NFATc2 might be linked to the pluripotency machinery through its regulatory action. Indeed, analysis of 4 lung AD cell lines showed transcripts of the major stemness factors *SOX2*, *OCT4* and *NANOG* were significantly elevated in tumorspheres compared to monolayers (*Figure 5—figure supplement 1*). Genetic inhibition of NFATc2 in HCC827 and PDCL#24 led to consistent *SOX2* repression with the highest magnitude of change compared to the other 2 factors (p<0.01) (*Figure 5A–B*), while all 3 were significantly upregulated on NFATc2 ectopic expression (*Figure 5C–D*). Corresponding changes were shown at the protein level (*Figure 5E*). Further, inhibition of calcineurin activity either by inhibitors (*Figure 5F–G*), or by siRNA transient knockdown of one of its subunits PPP3R1 (*Figure 5H–I*) consistently down-regulated SOX2 expression in both HCC827 and PDCL#24 cells. On the other hand, NFATc1 has been reported to be a transcriptional

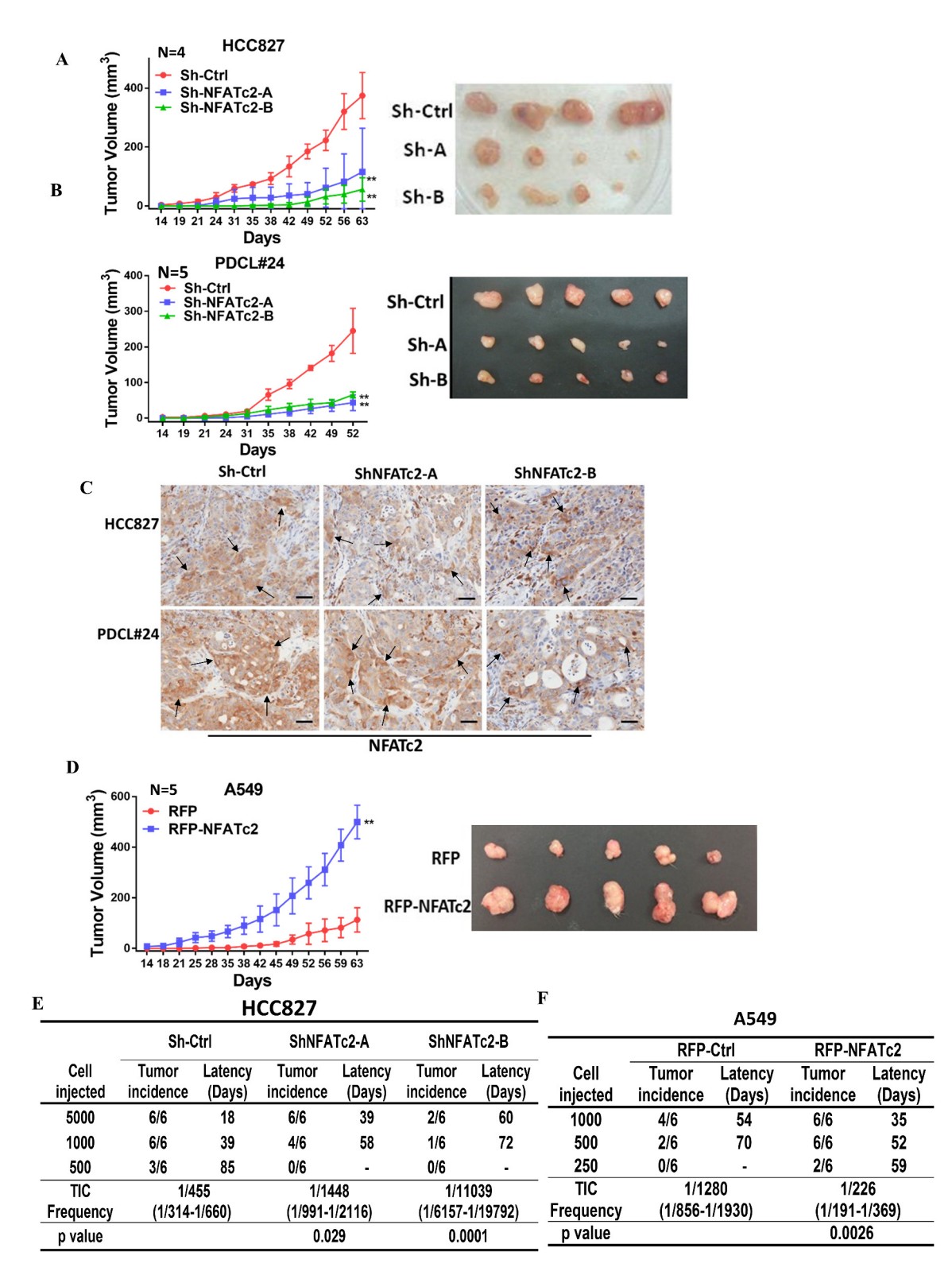

**Figure 3.** NFATc2 regulated tumorigenesis in vivo. (**A–B**) $1 \times 10^4$ of HCC827 cells (**A**), and PDCL#24 cells (**B**), respectively, were subcutaneously inoculated into the flanks of SCID mice, and tumor volumes were monitored. Representative tumor images and tumor growth curves are shown. **p<0.0001, comparison with respective control by two-way ANOVA. Error bar indicates the mean ±SD of tumor volumes of mice as indicated. (**C**) NFATc2 expression by IHC in xenografts generated from HCC827 or PDCL#24 cells, respectively, with or without NFATc2 knockdown. Tumor cells at

*Figure 3 continued on next page*

*Figure 3 continued*

the tumor/stroma interface (arrows) showed stronger NFATc2 expression, possibly due to micro-environmental induction. Scale bars, 50 µm. (D) $1 \times 10^4$ of A549 cells with or without NFATc2 overexpression were subcutaneously injected into SCID mice, and tumor volumes were monitored. Representative tumor images and tumor growth curves are shown. **p<0.0001, comparison with control by two-way ANOVA. Error bar indicates the mean ±SD of tumor volumes of mice as indicated. (E–F) Limiting dilution assay in vivo. Indicated numbers of HCC827 cells (E), and A549 cells (F) were subcutaneously inoculated into SCID mice, and the tumor incidence and latency were monitored for 3 months. The TIC frequency and P values were calculated using the L-Calc software (Stemcell Tech, Vancouver, Canada, http://www.stemcell.com).

The following source data and figure supplement are available for figure 3:

**Source data 1.** Statistical analyses for *Figure 3A,B and D*.
**Figure supplement 1.** NFATc2 regulated *in vivo* tumorigenesis.

regulator of SOX2 in pancreatic cancer (*Singh et al., 2015*). To study whether it is also involved in regulating SOX2, expression in lung AD, NFATc1 was transiently knocked down by siRNA in HCC827 and PDCL#24 cells, which did not result in consistently reduced SOX2 expression (*Figure 5J–K*), indicating NFATc2, rather than NFATc1, was involved in the regulation of SOX2 expression in lung AD. Together, the data suggested SOX2 is a major stemness target of the calcineurin/NFATc2 axis.

To further delineate the molecular mechanism of SOX2 regulation by NFATc2, we screened, in silico, the genomic sequences spanning 5 kb up- and downstream of the *SOX2* transcription start site (TSS), which identified 4 regions encompassing multiple conserved NFAT binding sequences (*Figure 6—figure supplement 1A*). Alignment with ChIP-seq data of A549 cells retrieved from public databases showed significant overlap at loci of H3K27Ac occupancy with regions 2 and 3, respectively (*Figure 6A*). Luciferase assays confirmed these regions are active transcriptional regulatory regions (*Figure 6—figure supplement 1B*). Further evaluation with respective SOX2 luciferase reporter revealed transcriptional activities were mediated by sites 1, 2, 4, and 5 (*Figure 6B*). Using H441 lung cancer cell line with NFATc2 transient overexpression, we observed only sites 1, 4 and 5 showed statistically significant increased reporter activities while those of sites 4 and 5 were reciprocally abolished by CSA treatment (*Figure 6C*). Finally, site directed mutagenesis of NFAT motifs (GGAAA to GACTA) prevented reporter activities of sites 4 and 5 only (*Figure 6D*), and the findings were supported by data from A549 and H1299 cells ectopically expressing NFATc2, respectively (*Figure 6—figure supplement 1C,D*). Notably, sequence homology analysis showed *SOX2* sites 4 and 5 are highly conserved across different mammalian species (*Figure 6E*). Thus, the data suggested NFATc2 was highly likely to regulate SOX2 expression through binding to 3' enhancers at sites 4 and 5. For validation, NFATc2 ChIP-qPCR assays were performed using A549 with NFATc2 upregulation, which showed statistically significant enrichment of sites 4 and 5 sequences compared to vector control (*Figure 6F*). In HCC827 cells, sites 4 and 5 sequences were significantly enriched by anti-NFATc2 antibody compared to IgG control. Conversely, these sequences were significantly reduced upon NFATc2 knockout in HCC827, compared to their endogenous levels in control cells, indicating de novo physical binding of NFATc2 to *SOX2* at sites 4 and 5 (*Figure 6G*). Together, the data showed NFATc2 upregulates SOX2 by binding to its 3' enhancer region at around 3.2 kb (site 4) and 3.6 kb (site 5) from the TSS, respectively.

## NFATc2 and SOX2 expressions were significantly correlated in human lung AD and NFATc2/SOX2 coupling augmented tumor functions

The clinical significance of NFATc2/SOX2 coupling was further assessed in human lung cancers. To avoid the confounding effect of *SOX2* gene amplification in squamous cell carcinoma (SCC), and to focus on tumors with demonstrated involvement of NFATc2, we performed IHC analysis on 92 moderately to poorly differentiated AD. A significant correlation between SOX2 and NFATc2 expressions was observed (*Figure 6H*). In lung AD cell lines, *NFATc2* and *SOX2* transcript expressions were also positively correlated (*Figure 6I*). Together, the data supported NFATc2 upregulates SOX2 in clinical and cultured lung AD.

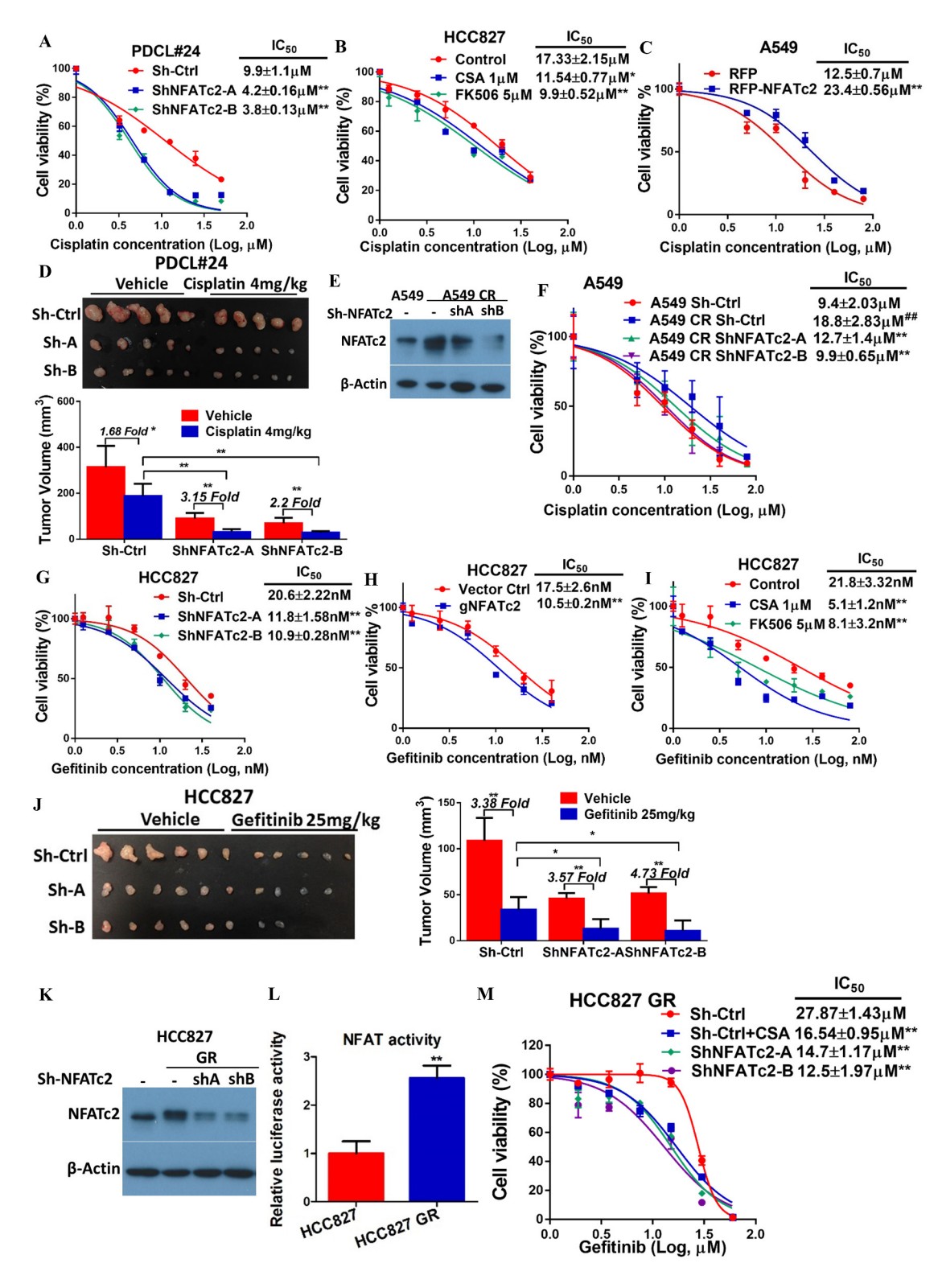

**Figure 4.** NFATc2 promoted resistance to cytotoxic and targeted therapy. (A) Effect of NFATc2 knockdown on cisplatin response of PDCL#24 cells by MTT assay. (B) Effect of CSA or FK506 treatment on cisplatin response of HCC827 cells by MTT assay. (C) Effect of NFATc2 overexpression on cisplatin response of A549 cells by MTT assay. *p<0.05, **p<0.01 versus control by t-test. (D) In vivo effect of NFATc2 knockdown on cisplatin response of PDCL#24 xenografts. 1 × 10[6] of PDCL#24 cells were subcutaneously inoculated into the flanks of Nude mice. Nude mice bearing subcutaneous

*Figure 4 continued on next page*

*Figure 4 continued*

xenografts were randomly separated into two groups and treated with intraperitoneal injections of cisplatin (4 mg/kg every three days) or saline control, respectively. Xenografts were photographed and histograms of tumor volumes were compared to vector and no-treatment controls. *p<0.05, **p<0. 01 by t-test. Error bar indicates the mean ±SD of tumor volumes of five mice. (E) NFATc2 expression by Western blot in A549 and corresponding cells with induced cisplatin-resistance (A549 CR) with or without NFATc2 knockdown. (F) Effects of NFATc2 knockdown on cisplatin sensitivity by MTT assays in A549 and A549 CR cells. ##p<0.01, versus vector control of parental cells, **p<0.01 versus CR Sh-Ctrl by t-test. (G–H) Dose response curves of gefitinib treatment by MTT assays of HCC827 cells with NFATc2 knockdown (G), or knockout (H). (I) Dose response curves of gefitinib treatment by MTT assays of HCC827 cells in the presence of CSA or FK506 for 72 hr. For G-I: **p<0.01 versus control by t-test. (J) Effects of NFATc2 stable knockdown on response of HC827 xenografts to gefitinib. $1 \times 10^6$ of HCC827 cells were subcutaneously inoculated into the flanks of Nude mice. Nude mice bearing subcutaneous xenografts were randomly separated into two groups and treated with gefitinib (25 mg/kg/day by oral gavage) or 1% Tween 80 as control. *p<0.05, **p<0. 01 by t-test. Error bar indicates the mean ±SD of tumor volumes of six mice. (K and L) NFATc2 expression with or without NFATc2 knockdown by Western blot (K), and NFAT activity by luciferase reporter assay (L), in HCC827 parental and gefitinib-resistant (GR) cells. (M) Gefitinib sensitivity of HCC827GR cells treated with CSA or with NFATc2 knockdown analyzed by MTT assays. **p<0.01 versus control by Student's t-test. For all MTT assays, error bar indicates mean ±SD for at least three replicates.

The following source data and figure supplements are available for figure 4:

**Source data 1.** Statistical analyses for *Figure 4D and J*, *Figure 4—figure supplement 1B* and *3A*.
**Figure supplement 1.** NFATc2 promoted cancer cell resistance to cisplatin treatment.
**Figure supplement 2.** NFATc2 promoted cancer cell resistance to paclitaxel treatment.
**Figure supplement 3.** NFATc2 promoted cancer cell resistance to gefitinib treatment.

Next, we evaluated whether NFATc2-induced SOX2 upregulation was functionally relevant for its role in sustaining TIC. Using A549 transduced for NFATc2 overexpression, SOX2 suppression by 2 shSOX2 sequences led to significantly reduced tumorspheres formation (*Figure 6J–K*), and cell motility (*Figure 6L*). In vivo, the xenografts enhanced by NFATc2 overexpression were also abrogated by SOX2 knockdown (*Figure 6M*).

Similar to NFATc2, SOX2 was also upregulated in A549 induced for cisplatin resistance (A549 CR) but on NFATc2 knockdown, SOX2 levels were repressed (*Figure 6N*), suggesting NFATc2/SOX2 coupling was functionally active in resistant cancer cells. Moreover, while NFATc2 overexpression induced cisplatin resistance of A549 cells, SOX2 silencing restored sensitivity to a level comparable to that of the control cells (*Figure 6L*). Overall, the data indicated NFATc2 induces TIC, cancer initiating phenotypes and drug resistance through upregulating SOX2 expression.

## ALDH1A1 was a target of NFATc2/SOX2 regulation

We have shown NFATc2 was upregulated in ALDH$^+$/CD44$^+$-TIC, suggesting NFATc2 might regulate this TIC population (*Figure 2B*). With NFATc2 knockdown or knockout in HCC827, ALDH$^+$/CD44$^+$-TIC were significantly reduced (*Figure 7A,B*). Consistent changes were observed for PDCL#24 with NFATc2 knockdown (*Figure 7—figure supplement 1A*). Further, inhibition of calcineurin by CSA and FK506, respectively, also significantly reduced ALDH$^+$/CD44$^+$-TIC (*Figure 7C*). Conversely, in both A549 with NFATc2 overexpression and in A549 CR cells, ALDH$^+$/CD44$^+$-TIC proportions were increased (*Figure 7D* and *Figure 7—figure supplement 1B*). Breakdown analysis showed the trend of changes that were more consistent with those of the ALDH$^+$ but not CD44$^+$ population, suggesting ALDH might be the main target of NFATc2.

ALDH1A1 is the most frequent and important ALDH isozyme reported in lung cancer TIC (*Ucar et al., 2009*; *Tomita et al., 2016*). Although ALDH1 is marketed as the major subtype contributing to ALDH activities detected by the ALDEFLUOR$^{TM}$ assay, cross-reactivity with other isoforms cannot be excluded. Hence, to explore the part contributed by ALDH1A1, we abrogated ALDH1A1 in A549 engineered to overexpress NFATc2, which led to significantly suppressed aldefluor activities (*Figure 7—figure supplement 2A*). *ALDH1A1* expression was suppressed in NFATc2 knockdown or knockout cells, as well as in cells treated with CSA or FK506, respectively (*Figure 7E–F*, *Figure 7—figure supplement 2B*). Conversely, *ALDH1A1* was up-regulated in NFATc2 overexpressing cells

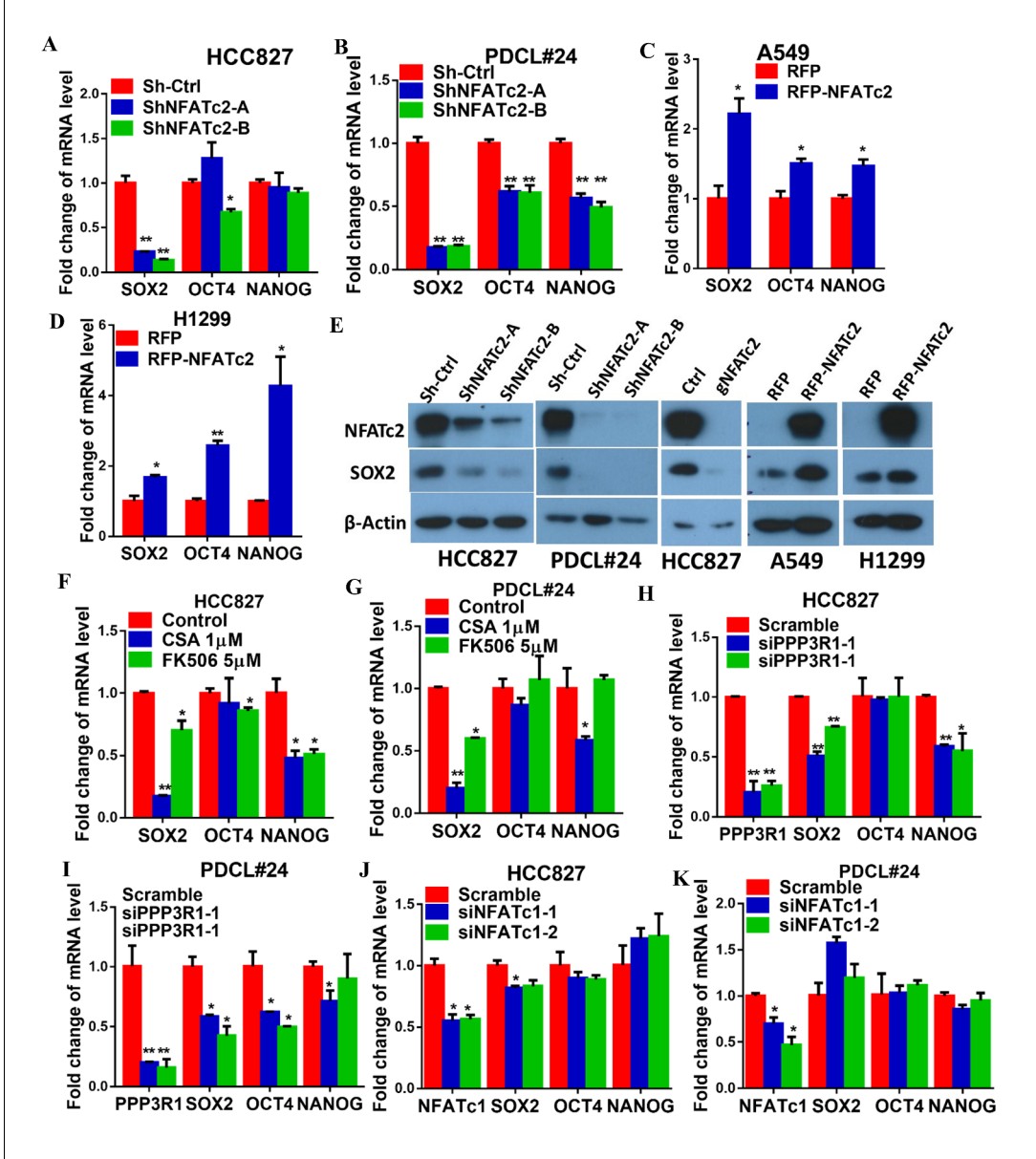

**Figure 5.** NFATc2 regulated SOX2 expression. (A–D) Pluripotency genes expressions analyzed by qPCR in HCC827 (A), PDCL#24 (B), A549 (C), and H1299 cells (D) with NFATc2 knockdown or overexpression. (E) Effects of stable NFATc2 knock-down, knockout or overexpression on SOX2 expression in respective lung cancer cells by Western blot analysis. (F–G) Pluripotency genes expression analyzed by qPCR in HCC827 (F), and PDCL#24 cells (G) treated with CSA or FK506, respectively, for 24 hr. (H–I) Effects of transient knockdown of PPP3R1 on pluripotency gene expressions analyzed by qPCR in HCC827 (H) and PDCL#24 cells (I). (J–K) Effects of transient knockdown of NFATc1 on pluripotency gene expressions analyzed by qPCR in HCC827 (J) and PDCL#24 cells (K). *p<0.05, **p<0.01 versus control by t-test. Error bar indicates the mean ±S.D. for at least three independent replicates.

The following figure supplement is available for figure 5:

**Figure supplement 1.** NFATc2 knockdown did not affect cell cycle progression of HCC827 cells.

and A549 CR cells (*Figure 7G–H*), indicating NFATc2 regulates *ALDH1A1* expression which contributes to the majority of ALDH positivity in ALDH$^+$/CD44$^+$-TIC.

Further analysis of the role of NFATc2/SOX2 coupling in ALDH1A1 regulation showed silencing SOX2 in NFATc2-overexpressing A549 cells consistently prevented the increase of ALDH$^+$ and ALDH$^+$/CD44$^+$ subpopulations only (*Figure 7I*). Specifically, the expected upregulation of *ALDH1A1*

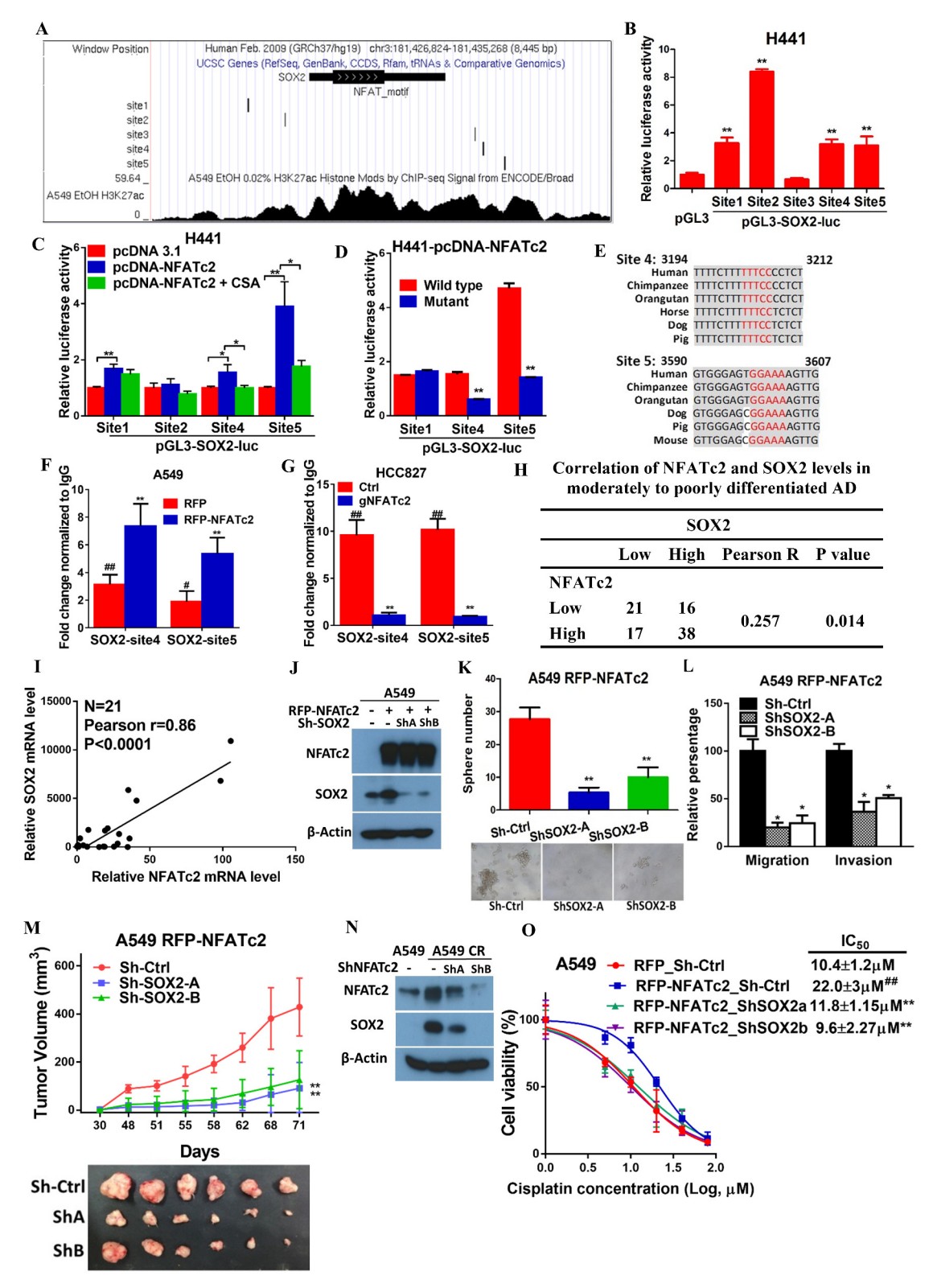

**Figure 6.** NFATc2 regulated tumor functions through trans-activating SOX2 expression. (**A**) Genome browser view of NFAT binding sites and H3K27Ac marks (lowest panel) on SOX2 regulatory regions (regions 2 and 3 indicated in *Figure 6—figure supplement 1*) analyzed in A549 cells. (**B**) Transcriptional activities of sites 1–5 by dual luciferase reporter assays in H441 cells. (**C**) Transcriptional activities of the indicated putative NFAT binding sites by respective luciferase reporters in H441 cells with transient NFATc2 over-expression, with or without CSA treatment. (**D**) Effects of site-directed
*Figure 6 continued on next page*

*Figure 6 continued*

mutagenesis of the indicated putative NFAT binding sequences by respective luciferase reporter assays in H441 cells with transient NFATc2 overexpression. For B-D, *p<0.05, **p<0.01 versus control by t-test. Error bar indicates the mean ±S.D. for at least three independent replicates. (E) Alignment of sites 4 and 5 genomic sequences showing highly homologous regions (gray) in different mammalian species, with putative NFAT binding sites highlighted in red. (F–G) ChIP–qPCR assays of NFATc2 binding to the indicated SOX2 sites in A549 cells with or without stable NFATc2 overexpression (F), or HCC827 cells with or without NFATc2 knockout (G). #p<0.05, ##p<0.01 versus IgG control,m **p<0.01 versus vector control by t-test. Error bar indicates the mean ±S.D. for at least three independent replicates. (H) Correlation of immunohistochemical expressions of NFATc2 and SOX2 in 92 moderately to poorly differentiated human lung adenocarcinoma by $\chi^2$-test. Pearson R, Pearson correlation coefficient. (I) Correlation of mRNA levels of SOX2 and NFATc2 in a panel of lung AD cell lines analyzed by q-PCR and Pearson correlation test. (J) Expression of NFATc2 and SOX2 in A549 cells with or without NFATc2 overexpression and SOX2 stable knockdown by Western blot. (K–L) Effect of SOX2 knockdown on tumorsphere formation (K), cell migration and invasion ability (L), of A549 cells with NFATc2 overexpression. *p<0.05, **p<0.01 versus control by t-test. Error bars indicate the mean ±SD for at least three independent replicates. (M) In vivo tumorigenicity of A549 cells with NFATc2 overexpression and SOX2 knockdown by subcutaneous inoculation of $1 \times 10^4$ cells in SCID mice. **p<0. 0001 versus control by two-way ANOVA. Error bar indicates the mean ±SD of tumor volumes of six mice. (N) Effect of NFATc2 knockdown on SOX2 expression in A549 CR cells analyzed by immunoblot. (L) Effect of SOX2 knockdown on cisplatin sensitivity by MTT assay of A549 cells with NFATc2 overexpression. ##p<0.01, versus vector control, **p<0.01 versus RFP-NFATc2_Sh-Ctrl by t-test. Error bar indicates the mean ±SD for at least three independent replicates.

The following source data and figure supplement are available for figure 6:

**Source data 1.** Statistical analyses for *Figure 6F and I*.
**Figure supplement 1.** NFATc2 regulated SOX2 expression through binding to 3' regulatory regions.

was also abolished (*Figure 7J*). In addition, as analyzed by IHC, xenografts derived from NFATc2-overexpressing A549 cells showed corresponding upregulation of SOX2 and ALDH1A1 (*Figure 7K*). Together, the data strongly supported NFATc2/SOX2/ALDH1A1 form a regulatory axis in lung cancer.

To examine the mechanism of SOX2 in ALDH1A1 regulation, ChIP-seq analysis was performed to identify SOX2 binding sequences in PDCL#24 cell line. Two loci of peak signals encompassing the SOX2 consensus binding motif (ATTCA) were identified in the 5' region of the *ALDH1A1* gene at around 27 kb from the TSS (site 1 and 2, *Figure 7L*). Bioinformatics analysis showed these loci were located at chr9:75,595,820–75,595,832 and chr9:75,602,860–75,602,872 which overlap with H3K27Ac occupancy deposited in public ChIP-seq database of A549 cells as well as mammalian conserved sequences (*Figure 7L*). On the other hand, no SOX2 binding motif was identified in regions flanking other *ALDH* isoform genes. Together, the findings suggested the presence of an accessible and highly conserved chromatin region encompassing putative SOX2-binding motifs 5' to the *ALDH1A1* gene. To confirm, ChIP-q-PCR assay using PDCL#24 cells was performed which showed anti-SOX2 antibodies significantly enriched both sequences 1 and 2 (*Figure 7M*). To study their transcriptional regulatory role, luciferase reporter constructs for sites 1 and 2 were co-transfected with SOX2 expression plasmids into A549 cells which yielded significantly enhanced reporter activities compared to control cells (*Figure 7N*). Compatible results were obtained for A549 with NFATc2 overexpression (*Figure 7—figure supplement 2C*).

To evaluate whether ALDH1A1 is a functionally relevant target, ALDH1A1 was knocked-down by siRNA in NFATc2-overexpressing A549 cells (*Figure 7—figure supplement 3*). This led to significant suppression of cell motility (*Figure 7O*) and cisplatin sensitization (*Figure 5N*). Furthermore, moderately to poorly differentiated human lung AD showed statistically significant positive correlation between SOX2 and ALDH1A1 expressions by IHC staining (*Figure 5O*). Collectively, the data supported ALDH1A1 is a functional target of regulation through NFATc2/SOX2 coupling.

## NFATc2/SOX2/ALDH1A1 coupling enhanced drug resistance and tumor properties through ROS attenuation

Alleviation of oxidative stress induced by chemotoxicity promotes cancer cell survival and mediates drug tolerance. Thus, in A549 CR cells, intracellular ROS levels were significantly lower compared to parental A549 cells (*Figure 8A*). To investigate for possible relation between NFATc2 and ROS modulation, NFATc2 was silenced by knockdown or knockout which led to increased ROS levels (*Figure 8B–D*). As shown in *Figure 8E*, NFATc2 depletion sensitized PDCL#24 cells to cisplatin

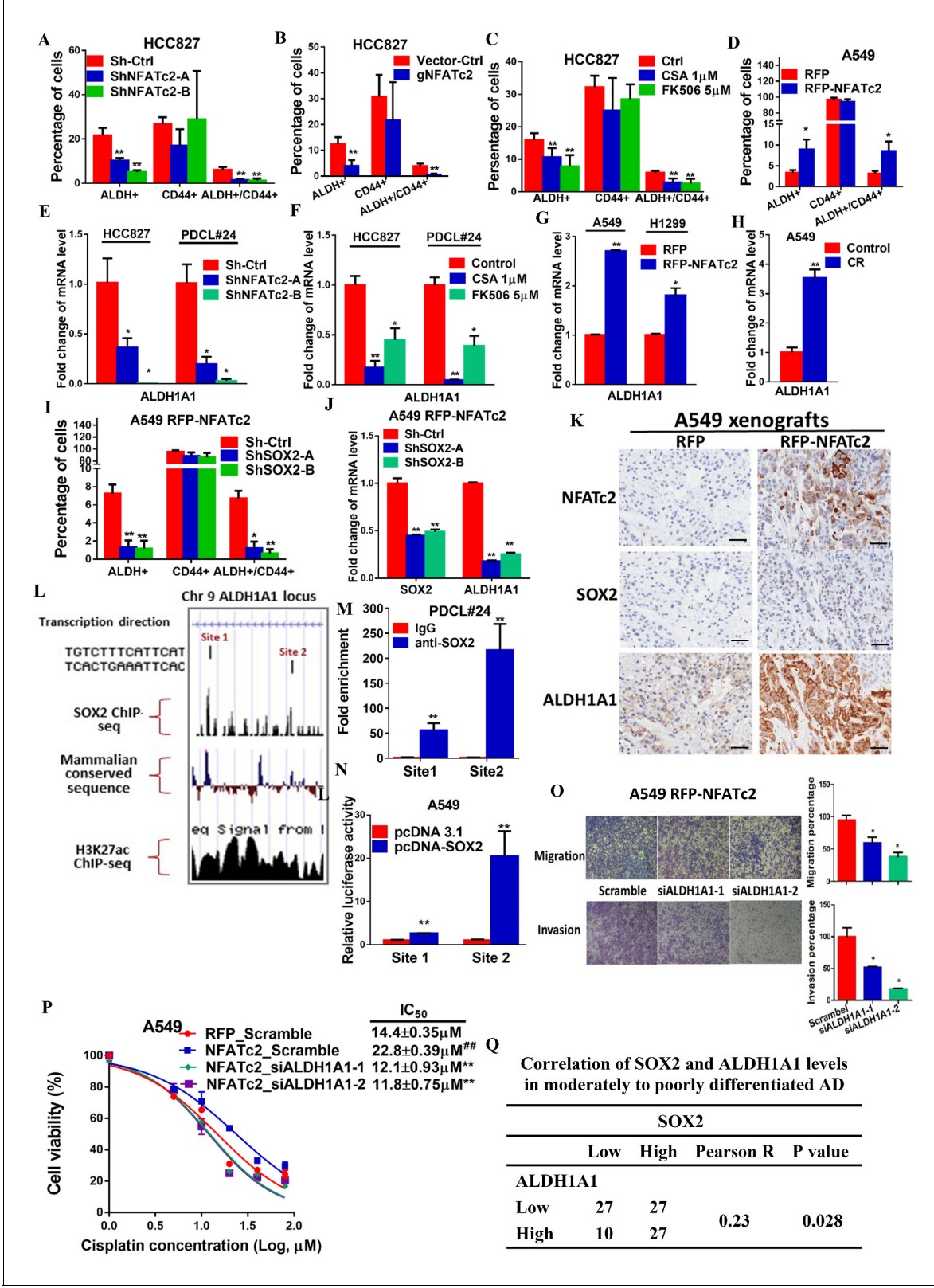

**Figure 7.** ALDH1A1 was a target of NFATc2/SOX2 coupling. (A–D) Effects on ALDH$^+$, CD44$^+$ and ALDH$^+$/CD44$^+$ cell populations by flow cytometry analysis of HCC827 with NFATc2 knockdown (A), NFATc2 knockout (B), or NFATc2 inhibition by CSA or FK506 (C), and of A549 cells with NFATc2 overexpression (D). (E–H) Effects on *ALDH1A1* mRNA expression by qPCR analysis of cancer cells with NFATc2 knockdown (E), NFATc2 inhibition by CSA or FK506 (F), NFATc2 up-regulation (G), or of A549 with induced cisplatin resistance (H). (I) Effects of SOX2 knockdown on ALDH$^+$, CD44$^+$ and

*Figure 7 continued on next page*

*Figure 7 continued*

ALDH$^+$/CD44$^+$ cell populations by flow cytometry in A549 with NFATc2-overexpression. (J) Expression of *SOX2* and *ALDH1A1* transcripts in A549 cells with NFATc2 overexpression and SOX2 knockdown. (K) Representative images of A549 xenografts with or without NFATc2 overexpression immunohistochemically stained for NFATc2, SOX2 and ALDH1A1, respectively. Scale bars, 50 μm. (L) Conserved SOX2 binding sequences (ATTCA) at ALDH1A1 enhancer region by ChIP-seq of PDCL#24 cells, aligned with homologous mammalian sequences and H3K27Ac peaks of A549 cells from published databases. (M) Detection of endogenous SOX2 binding to *ALDH1A1* sites by ChIP–qPCR analysis in PDCL#24 cells. (N) Luciferase reporter activities at sites 1 and 2 of *ALDH1A* enhancer region by dual luciferase reporter assay in A549 cells with SOX2 overexpression. (O–P) Effects of transient ALDH1A1 suppression on A549 with upregulated NFATc2 with respect to invasion and migration (O), and cisplatin sensitivity. ##$p<0.01$, versus RFP_Scramble by t-test. (P). (Q) Correlation between ALDH1A1 and SOX2 expressions by IHC in human lung adenocarcinomas by $\chi^2$-test. *$p<0.05$, **$p<0.01$ versus control by t-test. Error bar indicates the mean ±S.D. for at least three independent replicates.

The following source data and figure supplements are available for figure 7:

**Source data 1.** Statistical analyses for *Figure 7A,B,I and M*.
**Figure supplement 1.** NFATc2 regulated ALDH activity.
**Figure supplement 2.** NFATc2 regulated *ALDH1A1*.
**Figure supplement 3.** Effect of siALDH1A1 on ALDH1A1 expression.
**Figure supplement 4.** Effect of NFATc2/SOX2 on β-catenin activity.

treatment, but the addition of the reducing agent NAC reversed cisplatin IC$_{50}$ to above the control level dose-dependently. Reciprocally, the enhanced resistance of A549 by NFATc2 overexpression was reversed by oxidative stress induced by the glutathione inhibitor BSO (*Figure 8F*), consistent with the suggestion that drug resistance by NFATc2 is effected through ROS attenuation. Similarly, ROS regulation also supported other tumor phenotypes mediated by NFATc2. For example, tumorspheres suppression by NFATc2 knockdown was restored by NAC dose-dependently but in control cells, no significant changes were induced even in the presence of additional NAC (*Figure 8G–H*). Likewise, cell migration and invasion efficiencies inhibited by NFATc2 depletion were reversed by NAC (*Figure 6I*). To further address the involvement of SOX2 coupling and ALDH1A1, we showed suppression of ROS by NFATc2-overexpression in A549 cells were reversed by silencing SOX2 or ALDH1A1, respectively (*Figure 8J–K*). Together, the data suggested NFATc2/SOX2/ALDH1A1 form a functional axis in the homeostatic regulation of an optimal level of ROS for in vitro tumorigenicity, cell motility, and mediation of drug resistance.

## Discussion

The elucidation and disruption of TIC maintenance pathways offer the opportunity to eliminate the most resilient cancer cells and improve treatment outcome. Many studies have demonstrated cell populations expressing high levels of specific markers such as ALDH, CD44, CD166, CD133, etc. are enhanced for a multitude of tumor phenotypes, with tumorigenicity and drug resistance being clinically the most important. Constitutive stem cell programs or stress-induced pathways are the main TIC sustaining mechanisms but details of their regulation are still elusive. NFAT is a family of transcription factors with the calcium-responsive parlors NFATc1, -c2, -c3 and -c4 being expressed in a tissue-dependent manner. In this study, using NFATc2 depletion and overexpression models of multiple lung cancer cell lines in TIC-defining functional assays (*Pattabiraman and Weinberg, 2014*), as well as clinical evidence from excised human lung cancers, we showed NFATc2 mediates TIC phenotypes. In vitro cell renewability was demonstrated by tumorspheres passaged for consecutive generations and and in vivo tumorigenicity tumorigenicity was illustrated by the limiting dilution assay. In clinical tumors, high level NFATc2 segregated with impaired tumor differentiation, advanced pathological stage, shorter recurrence-free and overall survivals in NFATc2-positive NSCLC, suggesting NFATc2 mediates the more primitive and aggressive tumor phenotypes. In the literature, only one research group has reported NFATc2 expression in 52% of 159 lung cancers and similar to our findings, high expression was associated with late tumor stage and poor survival. In their studies,

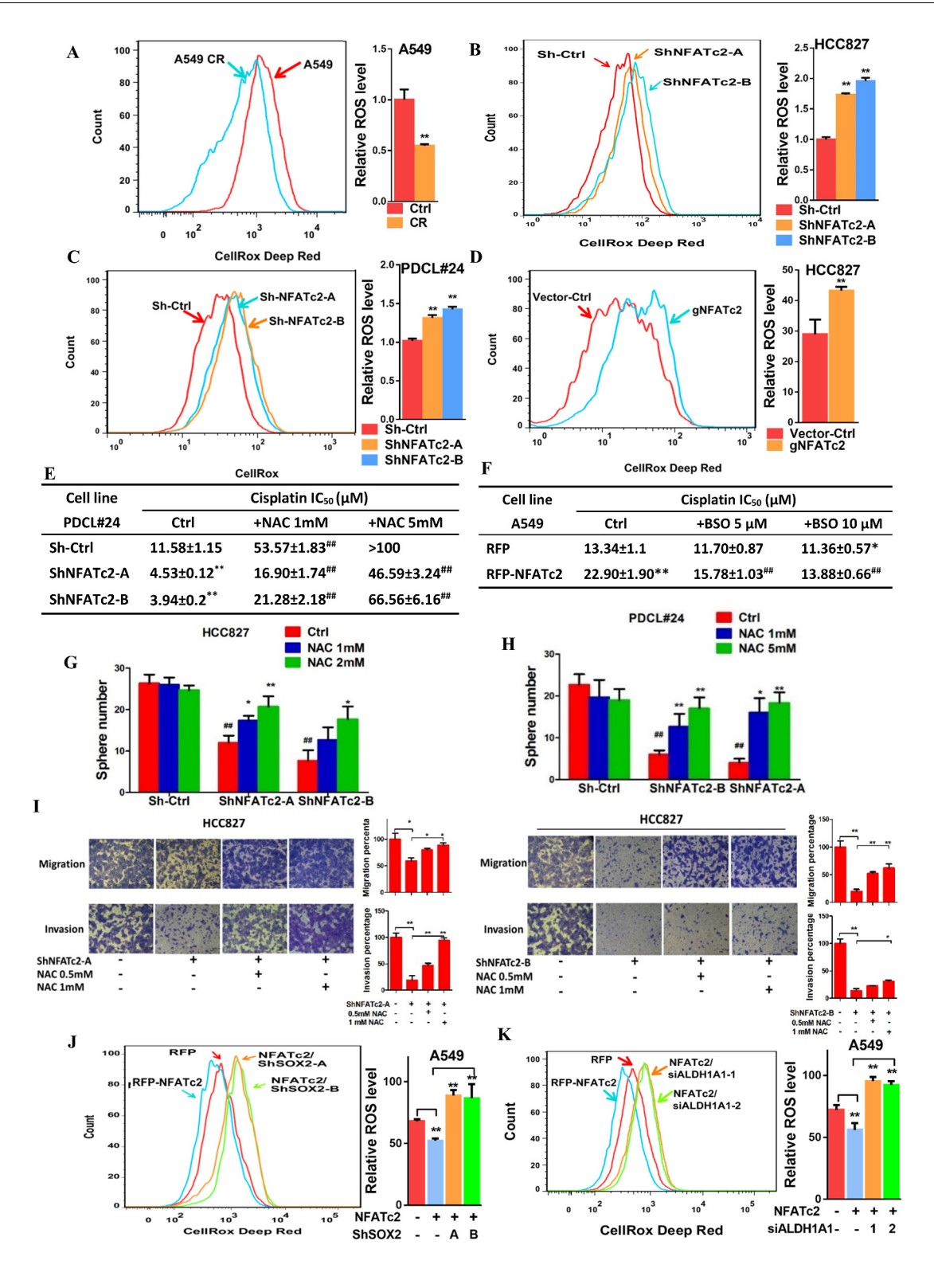

**Figure 8.** NFATc2 regulated TIC properties through ROS suppression. (**A**) ROS levels detected by flow cytometry in A549 and A549 CR cells. (**B–C**) ROS levels in HCC827 cells (**B**) and PDCL#24 cells (**C**) with or without NFATc2 stable knockdown. (**D**) ROS levels in HCC827 cells with or without NFATc2 knockout. (**E–F**) Cisplatin sensitivity expressed as IC$_{50}$ by MTT assays of NFATc2-silenced PDCL#24 cells treated with increasing doses of NAC (**E**), or NFATc2-overexpressing A549 cells treated with the oxidizing agent BSO (**F**), respectively. *p<0.05, **p<0.01 versus vector control without REDOX

*Figure 8 continued on next page*

*Figure 8 continued*

reagents; ##p<0.01 versus the corresponding treatment control by t-tests. Error bar indicates the mean ±S.D. for three independent replicates. (**G–H**) Effects of increasing doses of NAC on tumorsphere formation ability of HCC827 (**G**) cells and PDCL#24 cells (**H**). *p<0.05, **p<0.01 versus corresponding treatment controls, ##p<0.01 versus vector control by t-test. Error bar indicates the mean ±S.D. for three independent replicates. (**I**) Effects of increasing doses of NAC on cell migration and invasion of HCC827 cells with NFATc2 down-regulation by 2 sh-RNA knockdown sequences. (**J–K**) ROS levels in NFATc2-overexpressing A549 cells with stable SOX2 (**J**) or transient ALDH1A1 (**K**) knockdown. *p<0.05, **p<0.01 versus respective control by t-test. Error bar indicates the mean ±S.D. for at least three independent replicates.

The following source data is available for figure 8:

**Source data 1.** Statistical analyses for *Figure 8A–D,J and K*.

supportive evidences for cell proliferation, invasion and migration were demonstrated by cell models but the in vivo role of NFATc2 and, in particular, its effects on TIC, drug response or mechanisms of action were not addressed (*Chen et al., 2011*; *Liu et al., 2013b*). We have also evaluated public data sets on NFATc2 mRNA expression in lung cancer but a correlation with adverse patient outcome was not found (data not shown). Since NFATc2 is expressed by tumor infiltrating leukocytes, specific conclusions cannot be reached using this approach.

To identify the TIC sustaining mechanism of NFATc2, we hypothesized NFATc2 might be coupled to the core pluripotency factors SOX2, NANOG and/or OCT4, whose aberrant activities, if present, would be most suitable to orchestrate multifaceted cancer propensities through extensive transcriptional and epigenetic reprograming. Using multiple analyses of cancer cells and tumorspheres, we observed SOX2 was the most consistently altered factor with the highest magnitude of change when NFATc2 expression or calcineurin activity were manipulated. SOX2 is an important oncogene for squamous cell carcinomas (SCC) of the lung and other organs through SOX2 locus amplification at 3q26 (*Hussenet et al., 2010*; *Lu et al., 2010*; *Boumahdi et al., 2014*). For lung AD, although SOX2 is often expressed at high levels and predicts adverse survivals (*Chou et al., 2013*), distinct genetic mechanisms have not been identified, indicating regulatory or signaling aberrations might be involved. To evaluate the effects of NFATc2 on SOX2 expression, we avoided the potentially confounding element of SCC and focused on moderate to poorly differentiated AD where NFATc2 is shown to play an important prognostic role. Indeed, SOX2 expression is significantly correlated with that of NFATc2 in this group of human lung cancer. Functionally, NFATc2/SOX2 coupling contributes to tumor behavior as depletion of SOX2 in cell lines with NFATc2 overexpression led to significant suppression of TIC phenotypes. Hence, clinical and experimental evidences support NFATc2 impedes tumor differentiation and negatively affects patient outcome through coupling to SOX2 in lung AD.

We observed NFATc2 binds to the 3' enhancer region of *SOX2* at around 3.2 kb and 3.6 kb from TSS, effecting functional TIC enhancement and supporting the direct involvement of NFATc2 in stemness induction. In a recent study of a transgenic mouse model of pancreatic ductal adenocarcinoma with *KRAS G12VD* mutation and p53 heterozygous inactivation, Singh *et al* reported an analogous mechanism involving another NFAT family protein, NFATc1, which acts as a coactivator and transcriptional regulator of SOX2 (*Singh et al., 2015*). They showed NFATc1 played a permissive role for tumor dedifferentiation and expression of epithelial–mesenchymal transition (EMT) genes, while p53 disruption was essential for tumorigenesis, suggesting under the appropriate genetic context, NFATc1 activation transduces EMT through SOX2 upregulation. The authors proposed since chronic inflammation is a known etiology of pancreatic AD, NFATc1 might be a crucial factor involved in the progression of this cancer. On the other hand, we have shown NFATc1 inhibition does not result in SOX2 suppression, indicating NFATc2 is likely to be preferentially involved in lung cancers. Our data also distinguish a different SOX2 enhancer region accessible to NFATc2 which functions not only in the presence of *KRAS* mutation (A549, PDCL#24) but also in *EGFR* mutant (HCC827) cancers. Chronic inflammation is an important etiological mechanism of lung cancer through release of ROS and free radicals from alveolar macrophages and neutrophils. While many studies modeling carcinogenetic mechanisms of tobacco toxicity or chronic obstructive pulmonary diseases have featured the NF-κB pathway as a major mediator, our findings on the effects of NFATc2 on TIC induction might add to this repertoire. In fact, NFATc2 and NF-κB share highly

similar DNA binding domains but differ in the upstream activators, with NF-κB being stimulated by cytokine receptors and inflammatory molecules while NFATc2 is downstream of calcium signaling reputed as a stress response integrator, illustrating the multiple mechanisms through which inflammation-induced carcinogenesis might be initiated in the lung.

We have shown the ALDH[+]/CD44[+] fraction of lung cancer cells, despite being the smallest subset, demonstrates the highest tumorigenic capacity compared to counterpart subsets (*Liu et al., 2013a*). Evaluation of the role of NFATc2/SOX2 coupling in inducing this cell fraction revealed only ALDH but not CD44 showed consistent changes upon NFATc2 suppression or overexpression, respectively, and more specifically, ALDH1A1 was identified as a major functional target. Recent reports have shown β-catenin can directly regulate ALDH1A1 (*Condello et al., 2015*), and since SOX2 can upregulate β-catenin (*Yang et al., 2014*), this could infer SOX2 might indirectly regulate ALDH1A1 through β-catenin. We thus assessed β-catenin activity by western blot in A549 cells with NFATc2 overexpression and SOX2 knockdown, but no significant alternation of total or activated β-catenin levels was observed in these cells and the possible mechanism of indirect ALDH1A1 upregulation through β-catenin was not supported (*Figure 7—figure supplement 4*). On the other hand, computational screening did not detect significant and conserved SOX2-binding motifs within the proximal promoter region of ALDH1A1, but a more distant locus at 27 kb upstream from its TSS was shown to be a probable response region which we confirmed by CHIP-seq, CHIP-qPCR and luciferase reporter assays, respectively. This illustrates a distal enhancer is involved in the trans-regulation of ALDH1A1 expression, which has not been reported before. Current views on cancer stem cells suggest TIC is unlikely to be a single population with unique identifiers; instead, cell plasticity might induce variant TIC populations through dynamic response mechanisms, enabling cancer cells to meet the requirements of a complex micro-environment. In this connection, since we have only demonstrated the NFATc2/SOX2/ALDH1A1 axis, further investigation for the mechanisms of CD44 regulation in ALDH[+]/CD44[+]-TIC is needed.

Acquired drug resistance is mediated through complex genetic and molecular mechanisms (*Holohan et al., 2013*). We have shown NFATc2 augments cancer resistance with more prominent effects on tumors treated by cytotoxic chemotherapy. Response to targeted therapy is dominated by addiction to aberrant signaling from the mutant *EGFR*, and the effects of NFATc2 are modest by comparison. Nevertheless, as shown by the xenograft model of short term gefitinib treatment, complementary NFATc2 inhibition might be considered for patients receiving sub-therapeutic treatment regimes, e.g., due to the occurrence of serious skin or liver side effects.

Adaptive antioxidant response to alleviate oxidative stress from ROS surge during systemic therapy is one of the most important mechanisms of drug resistance (*Zhang et al., 2011*), suggested to be accentuated in cancer stem cells (*Diehn et al., 2009*; *Achuthan et al., 2011*; *Ishimoto et al., 2011*; *Chang et al., 2014*). Indeed, we have observed in multiple cell lines with induced resistance to chemotherapy or targeted therapy, NFATc2 was upregulated, while ROS was maintained at a lower level compared to parental cells. Changes in TIC phenotypes induced by NFATc2 up- or down-regulation were correspondingly restored by the redox reagents BSO or NAC, respectively, suggesting ROS scavenging is an important mechanism of drug resistance and other TIC properties mediated by NFATc2. In line with this suggestion, it has been reported in adult immortalized bronchial epithelial cells, NFAT can be upregulated in response to inflammatory and carcinogenic stimulation such as benzo-(a)-pyrene and heavy metals (*Huang et al., 2001*; *Ding et al., 2007*; *Cai et al., 2011*), leading to ROS-induced COX2 pathway signaling and enhanced cell survival (*Ding et al., 2006*). However, whether ROS is in turn suppressed by NFATc2 through a negative feedback mechanism has not been reported. Our findings supplement this information and show NFATc2 facilitates ROS scavenging, and further implicates this effect is mediated by ALDH1A1 through SOX2 coupling. This is consistent with findings of other studies on the role of ALDH1A1 in drug resistance through repressing ROS level (*Singh et al., 2013*; *Raha et al., 2014*; *Mizuno et al., 2015*).

In summary, this study demonstrates the calcium signaling molecule NFATc2 enhances functional characteristics associated with cancer stemness phenotype. Our data reveal a novel mechanism of SOX2 upregulation in lung cancers through enhancer binding by NFATc2. The NFATc2/SOX2/ALDH1A1 axis contributes to drug resistance by mediating a negative feedback mechanism for ROS scavenging and restoration of redox homeostasis. This study employed a candidate-based approach and the involvement of other potential NFATc2 and SOX2 targets in cancer phenotypes is not

addressed. Nevertheless, the findings implicate NFATc2 is a potential therapeutic target for sequential or combination therapy of lung cancer that aims to eliminate TIC.

## Materials and methods

### Cell lines

Established cell lines (H1993, HCC1833, H358, H1650, H2228, H1299, H1437, H1975, H23, H2122, HCC827, HCC78, A549, H441, and BEAS-2B) were obtained from ATCC. HCC366 and HCC78 were kindly provided by Dr. J. Minna (University of Texas Southwestern Medical Center. Dallas). All cell lines were kept as frozen aliquots upon receipt and only the first 20 passages were used in experiments. Patient-derived cell lines (HKULC1, HKULC2, HKULC3, HKULC4, PDCL#24, and FA31) were raised from resected primary lung cancers or malignant pleural effusions and only the 1st to 10th passages were used for study (*Lam et al., 2006*; *Liu et al., 2013a*). Cancer cells were maintained in RPMI-1640 (Invitrogen, Carlsbad, CA) with 10% FBS (Invitrogen, Carlsbad, CA). BEAS-2B were cultured in Keratinocyte-SFM (Invitrogen, Carlsbad, CA). Gefitinib, paclitaxel or cisplatin-resistant (-GR, TR or –CR, respectively) cells were generated by chronic exposure of cancer cells to stepwise increased doses of the respective drugs. All procured cell lines used in this study were free of mycoplasma contamination and were authenticated using the AmpFlSTR Identifiler PCR Amplification Kit for short tandem repeat profiling according to the manufacturer's instruction (Thermo Fisher Scientific, Waltham, MA). None of the cell lines used in this study were included in the list of commonly misidentified cell lines maintained by the International Cell Line Authentication Committee.

### SiRNA and plasmids

Small interfering RNA (siRNA) with pre-designed sequences targeting human NFATc1, PPP3R1, ALDH1A1 and scramble siRNA were from Sigma-Aldrich (St Louis, MO). pGL3-NFAT luciferase (17870), two shRNA sequences targeting SOX2, pLKO.1 Sox2 3HM a (26353) and pLKO.1 Sox2 3 hr b (26352), the negative control vector pLKO.1-puro (1864), the envelope vector pMD2.G (12259) and packaging vector psPAX2 (12260) were purchased from Addgene (Cambrige, MA; http://www.addgene.org). The pLKO.1-lentiviral shRNA with different inserts specifically targeting NFATc2 were purchased from Sigma-Aldrich (TRCN0000016144, TRCN0000230218). Human full length NFATc2 were amplified by PCR, and the RFP-NFTAc2 plasmids were generated by cloning the sequences into PCDH-CMV-MCS-EF1-COPRFP vector (SBI, Mountain View, CA). For luciferase reporter construction, SOX2 regulatory regions were amplified by PCR from human genomic DNA and cloned into pGL3 (Promega) to generate the SOX2-luc constructs. Primers used for genomic DNA amplification were listed in *Supplementary file 1A*. Site directed mutagenesis of the consensus NFAT binding site (GGAAA to GACTA) were performed using QuikChange (Stratagene).

### Lentiviral knockdown of NFATc2 and SOX2

Lentiviral shRNA was produced by transfecting the shRNA, envelope and packaging vectors into 293 T cells using lipofectamine 2000 (Invitrogen, Carlsbad, CA). Viruses were harvested after 48 hr of transfection followed by infection of target cells for 72 hr. Cells stably expressing shRNA were selected using puromycin (Sigma-Aldrich) for 14 days after 72 hr of viral infection.

### Lentiviral over-expression of NFATc2

RFP-NFATc2 lentiviral particles were produced and transduced into target cells using Lenti Starter kit (SBI, Mountain View, CA) according to manufacturer's instructions. RFP-positive cells stably over-expressing NFATc2 and SOX2 were selected by FACS using BD Aria (BD Biosciences).

### Lentiviral knock out of NFATc2 by CRISPR/Cas9

LentiCas9-Blast and lentiGuide-Puro were purchased from Addgene (Cambrige, MA; http://www.addgene.org). The gRNA targeting NFATc2 was designed using Zifit (http://zifit.partners.org/ZiFiT/) and listed in *Supplementary file 1A*. The annealed gNFATc2 oligonucleotides were cloned into lentiGuide-Puro. Lenti-viral cas9 and lenti-viral gNFATc2 were generated by transfecting lentiCas9-Blast or lenti-viral gNFATc2 together with pMD2.G and psPAX2, respectively, into 293FT cells by lipofectamine 2000 according to published protocols (*Sanjana et al., 2014*). After infection of lenti-viral

cas9, cells stably expressing Cas9 were selected using Blasticidin (Sigma-Aldrich) for 10 days. HCC827-Cas9 cells were further infected with lenti-gNFATc2 virus for 72 hr, and cells stably expressing gNFATc2 were selected using puromycin (Sigma-Aldrich) for 14 days.

## Flow cytometry and fluorescence activated cell sorting (FACS)

ALDH activity was analyzed by the Aldefluor kit (Stem Cell Technologies) according to manufacturer's instructions. CD44 expression was stained by anti-CD44-APC (BD Pharmingen) as previously described (*Liu et al., 2013a*). Flow cytometry was performed using FACS Canto II (BD Biosciences) and data were analyzed using FlowJo (Tree star). RFP positive cells with NFATc2 over-expression were isolated by FACS using BD Aria (BD Biosciences). Sorted cells were re-analyzed after collection to ensure a purity of >95%. Non-viable cells were identified by propidium iodide inclusion.

## Cell cycle analysis

Cells were harvested, washed once in PBS and fixed in 1 ml cold 70% ethanol for 1 hr at 4°C. The fixed cells were washed twice with PBS. Then 50 µl of RNase A solution (100 µg/ml) and 200 µl of propidium iodide (50 µg/ml) were added to the cell pellet. Cells were incubated at room temperature for 10 min. Fluorescence was measured by flow cytometry (FACS Canto II Analyzer, BD Biosciences) and data were analyzed using FlowJo (Tree star).

## BrdU cell proliferation assay

BrdU assay was performed using BrdU cell proliferation assay kit (Cell Signaling, Beverly, MA) according to manufacturer's instructions. Briefly, 5000 cells were seeded in a 96-well plate and incubated overnight followed by adding 10 µM BrdU and incubation for 12 hr. After the medium was removed, cells were fixed by 100 µl/well of fixing solution for 30 min at room temperature. BrdU was detected by 100 µl/well of 1X detection antibody solution followed by 1X HRP-conjugated secondary antibody solution. Then, 100 µl TMB substrate was added to each well and incubated at room temperature for 30 min followed by 100 µl of stop buffer. The absorbance was read at 450 nm using a plate spectrophotometer.

## Sphere formation and serial passage

Five hundred cells were seeded in an ultra-low plate (Costar) and cultured in cancer stem cell medium (RPMI-1640 medium supplemented with 20 ng/mL FGF, 20 ng/mL EGF, 40 ng/mL IGF and 1X B27 (Invitrogen, Carlsbad, CA) for 14 days. Tumorspheres were harvested, dissociated with trypsin, re-suspended in RPMI-1640, and 500 cells were seeded again for second passage using the same stem cell culture conditions.

## Cell motility assessment by migration and invasion assay

The migration and invasion assays were performed using Corning Transwell. Both chambers were filled with RMPI-1640 medium, and the lower chamber was supplemented with 10% FBS. For the migration assay, $5 \times 10^4$ cells were seeded into the upper chamber and allowed to migrate for 24 hr. For the invasion assay, the upper chamber was first coated with Matrigel (BD Pharmingen); $1 \times 10^5$ cells were seeded and allowed to invade for 24 to 36 hr. Cells that migrated or invaded to the lower surface of the transwells were fixed with methanol and stained with crystal violet. Cell densities were photographically captured in three random fields. The dye on the transwell membrane was dissolved by 10% acetic acid, transferred to a 96 well plate, and the dye intensity was measured by a plate spectrophotometer at 570 nm.

## Drug sensitivity assays

Drug sensitivity was tested by MTT assays. 6000 cells per well were seeded into 96-well plates and incubated for 24 hr at 37°C, followed by exposure to gefitinib (Selleckchem Houston, TX), paclitaxel (Sigma-Aldrich, St Louis, MO), or cisplatin (Sigma-Aldrich, St Louis, MO) at various concentrations for 72 hr with or without CSA (Selleckchem, Houston, TX), NAC or BSO (Sigma-Aldrich, St Louis, MO). Subsequently, Thiazolyl Blue Tetrazolium Bromide (MTT) (Sigma-Aldrich, St Louis, MO) was added and the mixture was incubated at 37°C for 4 hr. The absorbance was read at 570 nm using a

plate spectrophotometer. The drug response curve was plotted and $IC_{50}$ was calculated using non-linear regression model by GraphPad Prism 7.0.

## Quantitative PCR (qPCR) analysis

Total RNA was isolated using RNAiso Plus reagent (Takara, Mountain View, CA) and complementary DNA (cDNA) was generated using PrimeScript RT Reagent Kit (Takara, Mountain View, CA) according to the manufacturer's instructions. Gene mRNA levels were analyzed by quantitative RT-PCR (qPCR) (7900HT, Applied Biosystems, Carlsbad, CA) and SYBR green (Qiagen, Hilden, Germany) detection. Average expression levels of *RPL13A* and *beta-2-microglobulin* (*B2M*) were used as internal controls. Primers were listed in *Supplementary file 1B*.

## Western blot analysis

Cells were harvested and lysed on ice by lysis buffer [50 mM Tris HCl pH 7.4, 1% Triton X-100, 1 mM EDTA, 150 mM NaCl, 0.1% SDS, with freshly added 1:50 Phosphatase Inhibitor Cocktail 2 (Sigma), 1:50 Protease Inhibitor Cocktail (Sigma)] for 30 min. The cell lysate was then centrifuged at 13 k rpm for 20 min at 4°C to remove cell debris. The protein amount was quantified by the Dc Protein Assay (Bio-Rad). Cell lysates were resolved by 6–10% SDS-PAGE and then transferred onto PVDF membranes (Millipore). Primary antibodies including SOX2 (1:1000), NFATc2 (1:1000), $\beta$-catenin (1:1000), non p-$\beta$-catenin (1:1000), p- $\beta$catenin (1:1000) or ACTIN (1:1000) (Cell Signaling, Beverly, MA), respectively, where appropriate, were added. After overnight incubation, the membrane was washed with PBS and then incubated with the anti-rabbit secondary antibody. Target proteins on the membrane were visualized on X-ray films using ECL Plus Western Blotting Detection Reagents (Amersham, Buckinghamshire, UK).

## Chromatin immunoprecipitation (ChIP)-qPCR assay

ChIP assay was performed using the Magna ChIP™ A kit (Millipore, Billerica, MA) according to manufacturer's instructions. Briefly, cells were sonicated and lysed after protein/DNA cross-linking by 1% formaldehyde for 10 min. The crosslinked complex was immuno-precipitated by anti-NFATc2 antibody or control rabbit IgG (Cell Signaling, Beverly, MA) bound to protein A magnetic beads. After overnight incubation at 4°C, the complex was eluted and DNA was purified. The immune-precipitated DNA was quantified by qPCR using primer sequences designed to detect specific regulatory regions listed in *Supplementary file 1B*.

## ChIP-seq assay

ChIP assay was performed using the EZ-Magna ChIP A/G Chromatin Immunoprecipitation Kit (Millipore, 17–10086) according to manufacturer's instructions. Cells were cultivated and treated with 1% formaldehyde to crosslink protein and DNA. Cell lysate was sonicated to reduce the DNA length to 100 to 500 bp. The DNA-protein fragments were then incubated with 10 ug SOX2 antibodies (Abcam) and magnetic beads coated with protein A/G to form DNA-protein-antibody complex. The DNA was isolated and purified by Spin column and sent for commercial (BGI) library construction and sequencing using Illumina Hi-Seq platforms. Sequence reads were aligned to Human Reference Genome (hg19) using Bowtie (*Langmead et al., 2009*). Model-based analysis of ChIP-Seq (MACS) was used for peaks identification by comparing ChIP sample over input sample with default parameters (*Zhang et al., 2008*).

## NFATc2-binding sites predication

The 5'- and 3'- flanking regions (−5000 to +5000 bp) of SOX2 were scanned for NFAT binding sequences using PWMSCAN (*Levy and Hannenhalli, 2002*). The significance of the predicted sites was evaluated statistically using a permutation-based method and comparison with occurrence of the motif in background genomic sequences of intergenic regions. Phylogenetically non-conserved binding sites were filtered (*Li et al., 2010*).

## Luciferase reporter assay

Cells were transfected with luciferase reporters, expression plasmids and pRL-TK vector using lipofectamine 2000 (Invitrogen, Carlsbad, CA). Luciferase activities were measured by using the Dual-Luciferase Reporter Assay System (Promega).

## In vivo tumorigenicity

All animal experiments were performed after approval by the Animal Ethics Committee, the University of Hong Kong according to issued guidelines. Briefly, different numbers of cells mixed with an equal volume of matrigel (BD Pharmingen) were injected subcutaneously at the back of 6 week old severe combined immunodeficiency (SCID) mice or Ncr-nu/nu-nude mice. Tumor sizes were monitored every 3 days using digital vernier calipers, and tumor volumes were calculated using the formula [sagittal dimension (mm) $\times$ cross dimension (mm)$^2$]/2 and expressed in mm$^3$.

## Reactive oxygen species (ROS) measurement

Cells with or without respective treatments were washed with PBS and stained with 1 μM of the ROS probe CellROX$^{TM}$ Deep Red (Lift Technologies) for 30 mins according to manufacturer's instructions. Fluorescence was measured by flow cytometry (FACSCanto II Analyzer, BD Biosciences) and data were analyzed using FlowJo (Tree star).

## Human lung cancers

Surgically resected primary human NSCLC and corresponding normal lung tissues were collected prospectively in the Queen Mary Hospital, University of Hong Kong. Tissue collection protocols were approved by the Joint Hospital and University Institutional Review Board and written informed consents from patients were obtained. Fresh tissues were snap-frozen within 45–60 min after vascular clamping and kept in −70°C until use. Adjacent tumor tissues were fixed in 4% neural buffered formalin for 24 hr and processed into formalin fixed, paraffin embedded (FFPE) tissue blocks. Tumor classification and differentiation grading was according to the WHO classification of lung tumors, 2004. Tumor typing and pathological staging was performed by a qualified anatomical pathologist (MPW). Clinical parameters and outcomes were charted from hospital records in consultation with relevant clinicians.

## Immunohistochemistry (IHC)

Tissue microarrays were constructed using at least 5 cores of tissue from different representative tumor areas and 1 core of corresponding normal lung from each case. Tumor cores were randomly arranged in the microarray to prevent positional bias during recording of IHC results. De-paraffinized tissue microarray sections (5 μm) were subjected to antigen retrieval using microwave heating at 95°C in 1 mM EDTA buffer, pH 8.0. Endogenous peroxidase was quenched with 3% hydrogen peroxide for 10 min. Blocked sections were labeled with primary antibodies against NFATc2 (1:50 dilution, Cell Signaling), SOX2 (1:200 dilution, Cell Signaling) and ALDH1A1 (1:1000 dilution, Abcam) overnight at 4°C. Anti-rabbit HRP-labeled polymer (DAKO) was used as a secondary antibody. Color detection was performed by liquid DAB +substrate chromogen system (DAKO). Protein expression levels were semi-quantitatively analyzed using an automated image capturing and analysis system (Aperio).

NFATc2 expression level was scored according to the extent and intensity of nuclear staining in the tumor cells only and expression in the cytoplasm, stromal or inflammatory cells was excluded from evaluation. The intensity was graded as 1, 2, or 3 according to whether nuclear staining was absent or weak, moderate, or strong, respectively. The staining extent was graded as 1, 2, or 3 according to whether expression was observed in scattered individual cells, aggregates of 5 or more but <19 cells, or sheets of 20 or more cells. The products of the 2 grades were then computed, and cases with scores of 4 and above were counted as high level expression.

## Statistics

Data were analyzed by SPSS (version 16.0; SPSS Inc., Chicago, IL, USA), GraphPad Prism 7.0 or Excel (Microsoft, Redmond, WA, USA) software packages and shown as mean ±standard deviations (s.d.). Differential expression between paired tumor/normal tissues were analyzed by Wilcoxon text.

Differences between groups were analyzed by $t$ test for continuous variables. Differences between growth curves of xenograft model were analyzed by two-way ANOVA. Correlation between NFATc2 and SOX2 mRNA level were analyzed by Pearson correlation test. Correlation between NFATc2, SOX2, ALDH1A1 expressions and clinicopathological variables in lung cancers were analyzed by the $\chi^2$-test. Association between NFATc2 expression and overall survival and recurrence-free survival were analyzed by the Kaplan–Meier method with log-rank test. Multivariate survival analyses were performed by Cox regression model. Two-sided p values < 0.05 were considered as being statistically significant.

## Acknowledgement

We thank the Core Facility and Laboratory Animal Unit of the LKS Faculty of Medicine, The University of Hong Kong for technical support. We are thankful to Dr Terence Kin-Wah Lee, Dr. Stephanie Kwai-Yee Ma and Dr Judy Wai-Ping Yam for their helpful discussion on the study and comments on the manuscript.

## Additional information

### Funding

| Funder | Grant reference number | Author |
| --- | --- | --- |
| Research Grants Council, University Grants Committee | HKU 17123514 M | Zhi-Jie Xiao<br>Jing Liu<br>Si-Qi Wang<br>Yun Zhu<br>Xu-Yuan Gao<br>Vicky Pui-Chi Tin<br>Maria Pik Wong |
| University of Hong Kong | | Zhi-Jie Xiao<br>Jing Liu<br>Si-Qi Wang<br>Yun Zhu<br>Xu-Yuan Gao<br>Vicky Pui-Chi Tin<br>Maria Pik Wong |

The funders had no role in study design, data collection and interpretation, or the decision to submit the work for publication.

### Author contributions

Z-JX, Conceptualization, Data curation, Formal analysis, Validation, Investigation, Methodology, Writing—original draft, Writing—review and editing; JL, Conceptualization, Formal analysis, Investigation, Methodology; S-QW, Formal analysis, Investigation; YZ, Data curation, Software, Formal analysis, Investigation; X-YG, Data curation, Investigation; VP-CT, Data curation, Investigation, Methodology; JQ, Data curation, Software, Formal analysis; J-WW, Software, Supervision, Methodology; MPW, Conceptualization, Resources, Supervision, Funding acquisition, Writing—original draft, Project administration, Writing—review and editing

### Author ORCIDs

Zhi-Jie Xiao, http://orcid.org/0000-0001-6365-7278
Maria Pik Wong, http://orcid.org/0000-0003-4028-926X

### Ethics

Animal experimentation: All animal experiments were performed after approval by the Animal Ethics Committee, the University of Hong Kong according to issued guidelines. (CULATR No.4020-16)

## Additional files

Supplementary files
• Supplementary file 1.

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
