## [Decision Letter]

Thank you for submitting your article "NFATc2 enhances tumor-initiating phenotypes through the NFATc2/SOX2/ALDH axis in lung adenocarcinoma" for consideration by *eLife*. Your article has been favorably evaluated by Kevin Struhl (Senior Editor) and three reviewers, one of whom is a member of our Board of Reviewing Editors. The reviewers have opted to remain anonymous.

The reviewers have discussed the reviews with one another and the Reviewing Editor has drafted this decision to help you prepare a revised submission.

The authors have uncovered an interesting role for NFATc2 in lung cancer cell differentiation, growth and drug response. The authors found that in lung adenocarcinoma cell lines, the transcription factor NFATc2 promotes tumor initiating activities, migration, invasion, xenograft growth and resistance to cisplatin and EGFR inhibition. Mechanistically, NFATc2 seems to transactivate the pluripotency factor SOX2 through a 3' enhancer, and SOX2 is required for some of the biological activities of NFATC2 in this setting. Furthermore, NFATc2 and SOX2 exert some of their effects via transactivation of ALDH1A1 and attenuation of ROS production.

The reviewers discussed the findings and decided to encourage a resubmission that would address the following major concerns:

1) The majority of the studies use only two cell lines, rather than primary tumor cells, to make conclusions about TICs. The assays used in vivo are not selective for TIC activity, but could be a readout of tumor cell growth or proliferation. To meet the claims about TICs, data is needed showing that the effects of NFATc2 knockdown are not simply just an effect on proliferation. This can be done with a number of assays that measure cell cycle progression.

2) NFATc1 is mentioned in the Discussion but no data are shown. It needs to be determined if NFATc1 knockdown has the same effect as NFATc2.

3) Is calcium signaling responsible for changes in SOX2 expression changes? While genetic manipulation of NFATc2/SOX2 has been shown, does chemical inhibition of calcium signaling have the same effect? The nuclear translocation (and therefore activity) of the NFATc family of transcription factors is entirely dependent on the calcineurin phosphatase complex and there are potent and specific inhibitors of this complex. They only present a single experiment using the calcineurin inhibitor Cyclosporin A. In many experiments the use of CsA would add further strength to their arguments that the Cn/NFAT pathway is important for these phenotypes. Experiments with CsA should be added to at least some of their experiments even as a very simple control in their luciferase assays and added proof that this pathway is important in the responses to cisplatin and EGFR inhibitors. Does CsA or FK506 block sphere formation as is shown for NFATc2 knockdown in Figure 2 etc.?

4) If NFATc2 knock-down reduces tumor growth in the xenograft experiments, then it is predicted that the tumors that do grow from the shNFATc2 line would have selected away from knockdown and would express NFATc2. This should be investigated by qPCR, western, or IHC. This would further support the conclusion that it is knockdown of NFATc2 that leads to the reduction in tumor growth.

5) The comparisons in Figure 4 are not right. The effect of cisplatin on tumors size should be compared to the untreated samples with the same shRNA. So, its 1.68 fold reduction in sh-Cntrl but how much is sgNFATc1-A and -B reduced relative to the untreated tumors with that same shRNA? Then is that effect really any different than the response of the control tumors? This is really not right as it is shown now. It is not convincing that NFATc2 knock-down actually extenuates the response of tumors in vivo to the various treatments. The effects on the IC50 of the drug responses in vitro are also quite subtle and should be presented as such.

6) The conclusion that NFATc2 is a direct regulator of SOX2 expression is based in part on a ChIP experiment using artifact-prone protein overexpression. It would be important to demonstrate that 'endogenous' NFATc2 binds to the 3' enhancer in the SOX2 locus. The ChIP experiment in the paper is done in cells overexpressing RFP-NFATc2 and the data are not very convincing. The authors seem to count with antibodies that work well in Western blot and it would strengthen the paper to ChIP endogenous NFATc2. The authors also count with a pair of cell lines (A549 and A549CR) with vastly different levels of NFATc2, as well of isogenic NFATc2 KO cell lines, which would facilitate the experiment and ensure specificity in the signals.

---

## [Author Response]

*The reviewers discussed the findings and decided to encourage a resubmission that would address the following major concerns:*

*1) The majority of the studies use only two cell lines, rather than primary tumor cells, to make conclusions about TICs. The assays used in vivo are not selective for TIC activity, but could be a readout of tumor cell growth or proliferation. To meet the claims about TICs, data is needed showing that the effects of NFATc2 knockdown are not simply just an effect on proliferation. This can be done with a number of assays that measure cell cycle progression.*

To investigate whether the effects of NFATc2 could be mediated through promoting cell proliferation and cell cycle progression, we analyzed the proliferation index and cell cycle distribution of cancer cells with or without NFATc2 inhibition. Using the BrdU cell proliferation assay, we observed NFATc2 inhibition did not significantly reduce the proportion of proliferating cancer cells (Figure 2), while the ratios of cells in G0/G1, S, and G2/M phases were preserved (Figure 2). The findings do not support NFATc2 induces cell proliferation at the basal state, and thus this is unlikely to be the sole mechanism for its effects on TIC. These results are included in the last paragraph of the subsection “NFATc2 was overexpressed in lung TIC and mediated TIC properties”.

*2) NFATc1 is mentioned in the Discussion but no data are shown. It needs to be determined if NFATc1 knockdown has the same effect as NFATc2.*

In the Discussion, we referred to the transgenic mouse model of pancreatic ductal adenocarcinoma by Singh et al., in which carcinogenesis was driven by *KRAS^G12VD^* and *TP53* heterozygous inactivation. NFATc1 was found to mediate cancer functions by transactivating SOX2 through a 3’ enhancer.

As suggested by the reviewers, we investigated whether NFATc1 might also be involved in lung adenocarcinomas. Unlike the effects of inhibiting NFATc2, transient NFATc1 suppression using si-RNA did not lead to significant reduction of *SOX2* transcripts in HCC827 or PDCL#24 cells (Figure 5), implicating NFATc1 and NFATc2 are likely to be involved in cancer functions in a tissue-specific and/or context dependent manner. These results are described in the first paragraph of the subsection “NFATc2 upregulated SOX2 expression through its 3’ enhancer” and discussed in the third paragraph of the Discussion.

*3) Is calcium signaling responsible for changes in SOX2 expression changes? While genetic manipulation of NFATc2/SOX2 has been shown, does chemical inhibition of calcium signaling have the same effect? The nuclear translocation (and therefore activity) of the NFATc family of transcription factors is entirely dependent on the calcineurin phosphatase complex and there are potent and specific inhibitors of this complex. They only present a single experiment using the calcineurin inhibitor Cyclosporin A. In many experiments the use of CsA would add further strength to their arguments that the Cn/NFAT pathway is important for these phenotypes. Experiments with CsA should be added to at least some of their experiments even as a very simple control in their luciferase assays and added proof that this pathway is important in the responses to cisplatin and EGFR inhibitors. Does CsA or FK506 block sphere formation as is shown for NFATc2 knockdown in Figure 2 etc.?*

Thank you for the reviewers’ suggestions. We have performed additional experiments using both CSA and FK506 to inhibit de-phosphorylation and activation of NFAT through the calcineurin/NFAT complex, including effects on sphere formation (Figure 2), cisplatin and gefitinib sensitivity (Figure 4), pluripotency factors expression (Figure 5), luciferase activity (Figure 6), ALDH/CD44 distribution (Figure 7), and ALDH1A1 expression (Figure 7).

The effects of CSA and FK506 were similar to those of NFATc2 suppression. Inhibition of calcineurin led to suppression of sphere formation, sensitization of cells to cisplatin and gefitinib, reduced SOX2 expression and transcriptional activities of luc-SOX2-site 4 and 5. More specifically, we also transiently knocked down PPP3R1, one of the subunits of calcineurin, by two siRNA which led to reduction of SOX2 expression (Figure 5).

*4) If NFATc2 knock-down reduces tumor growth in the xenograft experiments, then it is predicted that the tumors that do grow from the shNFATc2 line would have selected away from knockdown and would express NFATc2. This should be investigated by qPCR, western, or IHC. This would further support the conclusion that it is knockdown of NFATc2 that leads to the reduction in tumor growth.*

We agree with the reviewers’ views. Using immunohistochemistry, we observed NFATc2 expression in a subpopulation of xenograft tumor cells, indicating tumor cells that had selected away from knockdown contributed to xenograft growth. Moreover, in many areas, NFATc2 expression was more prominent in tumor cells at the tumor/stroma interface where exogenous stimulation from the micro-environment, e.g. through the Wnt/Ca signaling pathway, might take place. This is in contrast to the control xenografts where NFATc2 expression was more diffuse throughout thick trabeculae of tumor cells. These results were shown in Figure 3 and described in the subsection “NFATc2 mediated tumorigenesis in vivo”, and in Figure 3 legend.

*5) The comparisons in Figure 3 are not right. The effect of cisplatin on tumors size should be compared to the untreated samples with the same shRNA. So, its 1.68 fold reduction in sh-Cntrl but how much is sgNFATc1-A and -B reduced relative to the untreated tumors with that same shRNA? Then is that effect really any different than the response of the control tumors? This is really not right as it is shown now. It is not convincing that NFATc2 knock-down actually extenuates the response of tumors in vivo to the various treatments. The effects on the IC50 of the drug responses in vitro are also quite subtle and should be presented as such.*

Thank you for the reviewers’ comments. We have changed the comparison groupings as suggested. As shown in Figure 4, both groups of NFATc2 knockdown mice (shNFATc2-A and shNFATc2-B) showed accentuated responses to cisplatin treatment, resulting in 3.15 and 2.2 folds tumor shrinkage, respectively, while the control xenografts shrunk by 1.68 fold only. From the tumor growth curve (Figure 4—figure supplement 1), tumor growth in all cisplatin treated groups were significantly slower than their respective vehicle control groups. Amongst the treated groups, the growth of tumors with NFATc2 knockdown was also significantly slower than those without knockdown. Together, the data indicated NFATc2 knockdown mediated additive effects and increased cisplatin sensitivity of the lung cancer cells evaluated.

For gefitinib response, as shown in Figure 4, although both groups of NFATc2 knockdown mice resulted in higher folds of tumor shrinkage (3.57 fold, 4.73 fold, respectively) compared to scramble control (3.38 fold), the added effects of NFATc2 inhibition were very modest. This is due to oncogene addiction of HCC827 to mutant EGFR which causes precipitous tumor shrinkage upon gefitinib exposure. Nevertheless, tumors harvested from the gefitinib treated NFATc2 knockdown groups were significantly smaller than those from gefitinib treated Sh-Ctrl group (Figure 4).

Using an alternative model, we investigated the effects of NFATc2 inhibition on response to short term gefitinib treatment for 5 days to allow the tumors to regrow. In mice with NFATc2 knockdown, tumor regrowth was observed in only 3 and 2 mice of the sh-NFATc2-A and sh-NFATc2-B groups, respectively. In contrast, all mice in the control group showed tumor regrowth. The NFATc2-inhibited xenografts showed much more pronounced inhibition of tumor resurgence (5.44 fold, 39.73 fold, respectively) compared to the control group (3.54 fold) and differences in tumor volumes on the growth curve were statistically significant by 2-way ANOVA (Figure 4—figure supplement 3).

The data provided additional evidence that NFATc2 has a tumor-supportive effect and increases tumor resistance. Clinically, patients receiving tyrosine kinase inhibitors for *EGFR* mutated lung adenocarcinomas often experience skin and liver side effects requiring reduced drug dosage or increased administration intervals. The addition of NFATc2 inhibitors might be beneficial as a complementary treatment. These results were described in the second paragraph of the subsection “NFATc2 promoted cancer resistance to cytotoxic and targeted therapy” and discussed in the fifth paragraph of the Discussion.

For the in vitro MTT assays, results of all treatment groups were normalized to their respective untreated controls when the IC_50_ was calculated, i.e. the effects of NFATc2 knockdown on cell viability have been normalized and the differences in IC_50_ were entirely due to drug sensitization effects. We apologize for the confusing use of the terms “significantly increased/reduced”. As suggested by the reviewers, we have omitted these and just stated the p value, or replaced them with “statistically significant”.

*6) The conclusion that NFATc2 is a direct regulator of SOX2 expression is based in part on a ChIP experiment using artifact-prone protein overexpression. It would be important to demonstrate that 'endogenous' NFATc2 binds to the 3' enhancer in the SOX2 locus. The ChIP experiment in the paper is done in cells overexpressing RFP-NFATc2 and the data are not very convincing. The authors seem to count with antibodies that work well in Western blot and it would strengthen the paper to ChIP endogenous NFATc2. The authors also count with a pair of cell lines (A549 and A549CR) with vastly different levels of NFATc2, as well of isogenic NFATc2 KO cell lines, which would facilitate the experiment and ensure specificity in the signals.*

Thank you for the suggestions. We have performed ChIP-qPCR experiments using HCC827 without NFATc2 manipulation and results showed endogenous NFATc2 binding to SOX2 enhancers. With NFATc2 knockout, markedly reduced levels were observed. The results were shown in Figure 5.